# Graph Condensation for Graph Neural Networks

**Wei Jin** [*]
Michigan State University
`jinwei2@msu.edu`

**Lingxiao Zhao**
Carnegie Mellon University
`lingxiao@cmu.edu`

**Shichang Zhang**
UCLA
`shichang@cs.ucla.edu`

**Yozen Liu**
Snap Inc.
`yliu2@snap.com`

**Jiliang Tang**
Michigan State University
`tangjili@msu.edu`

**Neil Shah**
Snap Inc.
`nshah@snap.com`

## Abstract

Given the prevalence of large-scale graphs in real-world applications, the storage and time for training neural models have raised increasing concerns. To alleviate the concerns, we propose and study the problem of *graph condensation* for graph neural networks (GNNs). Specifically, we aim to condense the large, original graph into a small, synthetic and highly-informative graph, such that GNNs trained on the small graph and large graph have comparable performance. We approach the condensation problem by imitating the GNN training trajectory on the original graph through the optimization of a gradient matching loss and design a strategy to condense node features and structural information simultaneously. Extensive experiments have demonstrated the effectiveness of the proposed framework in condensing different graph datasets into informative smaller graphs. In particular, we are able to approximate the original test accuracy by 95.3% on Reddit, 99.8% on Flickr and 99.0% on Citeseer, while reducing their graph size by more than 99.9%, and the condensed graphs can be used to train various GNN architectures. Code is released at https://github.com/ChandlerBang/GCond.

## 1 Introduction

Many real-world data can be naturally represented as graphs such as social networks, chemical molecules, transportation networks, and recommender systems (Battaglia et al., 2018; Wu et al., 2019b; Zhou et al., 2018). As a generalization of deep neural networks for graph-structured data, graph neural networks (GNNs) have achieved great success in capturing the abundant information residing in graphs and tackle various graph-related applications (Wu et al., 2019b; Zhou et al., 2018).

However, the prevalence of large-scale graphs in real-world scenarios, often on the scale of millions of nodes and edges, poses significant computational challenges for training GNNs. More dramatically, the computational cost continues to increase when we need to retrain the models multiple times, e.g., under incremental learning settings, hyperparameter and neural architecture search. To address this challenge, a natural idea is to properly simplify, or reduce the graph so that we can not only speed up graph algorithms (including GNNs) but also facilitate storage, visualization and retrieval for associated graph data analysis tasks.

There are two main strategies to simplify graphs: graph sparsification (Peleg & Schäffer, 1989; Spielman & Teng, 2011) and graph coarsening (Loukas & Vandergheynst, 2018; Loukas, 2019) . Graph sparsification approximates a graph with a sparse graph by reducing the number of edges, while graph coarsening directly reduces the number of nodes by replacing the original node set with its subset. However, these methods have some shortcomings: (1) sparsification becomes much less promising in simplifying graphs when nodes are also associated with attributes as sparsification does not reduce the node attributes; (2) the goal of sparsification and coarsening is to preserve some graph

---

[*] Work done while author was on internship at Snap Inc.

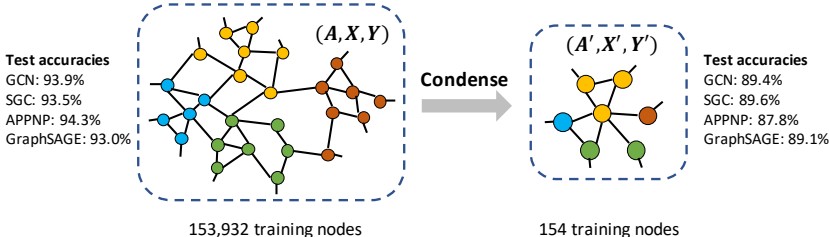

Figure 1: We study the graph condensation problem, which seeks to learn a small, synthetic graph, features and labels $\{\mathbf{A}', \mathbf{X}', \mathbf{Y}'\}$ from a large, original dataset $\{\mathbf{A}, \mathbf{X}, \mathbf{Y}\}$, which can be used to train GNN models that generalize comparably to the original. **Shown:** An illustration of our proposed GCOND graph condensation approach's empirical performance, which exhibits *95.3%* of original graph test performance with *99.9%* data reduction.

properties such as principle eigenvalues (Loukas & Vandergheynst, 2018) that could be not optimal for the downstream performance of GNNs. In this work, we ask if it is possible to significantly reduce the graph size while providing sufficient information to well train GNN models.

Motivated by dataset distillation (Wang et al., 2018) and dataset condensation (Zhao et al., 2021) which generate a small set of images to train deep neural networks on the downstream task, we aim to condense a given graph through learning a synthetic graph structure and node attributes. Correspondingly, we propose the task of *graph condensation*[1]. It aims to minimize the performance gap between GNN models trained on a synthetic, simplified graph and the original training graph. In this work, we focus on attributed graphs and the node classification task. We show that we are able to reduce the number of graph nodes to as low as 0.1% while training various GNN architectures to reach surprisingly good performance on the synthetic graph. For example, in Figure 1, we condense the graph of the Reddit dataset with 153,932 training nodes into only 154 synthetic nodes together with their connections. In essence, we face two challenges for graph condensation: (1) how to formulate the objective for graph condensation tractable for learning; and (2) how to parameterize the to-be-learned node features and graph structure. To address the above challenges, we adapt the gradient matching scheme in (Zhao et al., 2021) and match the gradients of GNN parameters w.r.t. the condensed graph and original graph. In this way, the GNN trained on condensed graph can mimic the training trajectory of that on real data. Further, we carefully design the strategy for parametrizations for the condensed graph. In particular, we introduce the strategy of parameterizing the condensed features as free parameters and model the synthetic graph structure as a function of features, which takes advantage of the implicit relationship between structure and node features, consumes less number of parameters and offers better performance.

Our contributions can be summarized as follows:

1. We make the first attempt to condense a large-real graph into a small-synthetic graph, such that the GNN models trained on the large graph and small graph have comparable performance. We introduce a proposed framework for graph condensation (GCOND) which parameterizes the condensed graph structure as a function of condensed node features, and leverages a gradient matching loss as the condensation objective.
2. Through extensive experimentation, we show that GCOND is able to condense different graph datasets and achieve comparable performance to their larger counterparts. For instance, GCOND approximates the original test accuracy by 95.3% on Reddit, 99.8% on Flickr and 99.0% on Citeseer, while reducing their graph size by more than 99.9%. Our approach consistently outperforms coarsening, coreset and dataset condensation baselines.
3. We show that the condensed graphs can generalize well to different GNN test models. Additionally, we observed reliable correlation of performances between condensed dataset training and whole-dataset training in the neural architecture search (NAS) experiments.

## 2 RELATED WORK

**Dataset Distillation & Condensation.** Dataset distillation (DD) (Wang et al., 2018; Bohdal et al., 2020; Nguyen et al., 2021) aims to distill knowledge of a large training dataset into a small synthetic

---

[1]We aim to condense both graph structure and node attributes. A formal definition is given in Section 3.

dataset, such that a model trained on the synthetic set is able to obtain the comparable performance to that of a model trained on the original dataset. To improve the efficiency of DD, dataset condensation (DC) (Zhao et al., 2021; Zhao & Bilen, 2021) is proposed to learn the small synthetic dataset by matching the gradients of the network parameters w.r.t. large-real and small-synthetic training data. However, these methods are designed exclusively for image data and are not applicable to non-Euclidean graph-structured data where samples (nodes) are interdependent. In this work, we generalize the problem of dataset condensation to graph domain and we seek to jointly learn the synthetic node features as well as graph structure. Additionally, our work relates to coreset methods (Welling, 2009; Sener & Savarese, 2018; Rebuffi et al., 2017), which seek to find informative samples from the original datasets. However, they rely on the presence of representative samples, and tend to give suboptimal performance.

**Graph Sparsification & Coarsening.** Graph sparsification and coarsening are two means of reducing the size of a graph. Sparsification reduces the number of edges while approximating pairwise distances (Peleg & Schäffer, 1989), cuts (Karger, 1999) or eigenvalues (Spielman & Teng, 2011) while coarsening reduces the number of nodes with similar constraints (Loukas & Vandergheynst, 2018; Loukas, 2019; Deng et al., 2020), typically by grouping original nodes into super-nodes, and defining their connections. Cai et al. (2021) proposes a GNN-based framework to learn these connections to improve coarsening quality. Huang et al. (2021b) adopts coarsening as a preprocessing method to help scale up GNNs. Graph condensation also aims to reduce the number of nodes, but aims to learn synthetic nodes and connections in a supervised way, rather than unsupervised grouping as in these prior works. Graph pooling is also related to our work, but it targets at improving graph-level representation learning (see Appendix D).

**Graph Neural Networks.** Graph neural networks (GNNs) are a modern way to capture the intuition that inferences for individual samples (nodes) can be enhanced by utilizing graph-based information from neighboring nodes (Kipf & Welling, 2017; Hamilton et al., 2017; Klicpera et al., 2019; Velickovic et al., 2018; Wu et al., 2019b;a; Liu et al., 2020; 2021; You et al., 2021; Zhou et al., 2021; Zhao et al., 2022). Due to their prevalence, various real-world applications have been tremendously facilitated including recommender systems (Ying et al., 2018a; Fan et al., 2019), computer vision (Li et al., 2019) and drug discovery (Duvenaud et al., 2015).

**Graph Structure Learning.** Our work is also related to graph structure learning, which explores methods to learn graphs from data. One line of work (Dong et al., 2016; Egilmez et al., 2017) learns graphs under certain structural constraints (e.g. sparsity) based on graph signal processing. Recent efforts aim to learn graphs by leveraging GNNs (Franceschi et al., 2019; Jin et al., 2020; Chen et al., 2020). However, these methods are incapable of learning graphs with smaller size, and are thus not applicable for graph condensation.

## 3 METHODOLOGY

In this section, we present our proposed *graph condensation* framework, GCOND. Consider that we have a graph dataset $\mathcal{T} = \{\mathbf{A}, \mathbf{X}, \mathbf{Y}\}$, where $\mathbf{A} \in \mathbb{R}^{N \times N}$ is the adjacency matrix, $N$ is the number of nodes, $\mathbf{X} \in \mathbb{R}^{N \times d}$ is the $d$-dimensional node feature matrix and $\mathbf{Y} \in \{0, \dots, C-1\}^N$ denotes the node labels over $C$ classes. Graph condensation aims to learn a small, synthetic graph dataset $\mathcal{S} = \{\mathbf{A}', \mathbf{X}', \mathbf{Y}'\}$ with $\mathbf{A}' \in \mathbb{R}^{N' \times N'}$, $\mathbf{X}' \in \mathbb{R}^{N' \times D}$, $\mathbf{Y}' \in \{0, \dots, C-1\}^{N'}$ and $N' \ll N$, such that a GNN trained on $\mathcal{S}$ can achieve comparable performance to one trained on the much larger $\mathcal{T}$. Thus, the objective can be formulated as the following bi-level problem,

$$\min_{\mathcal{S}} \mathcal{L}\left(\text{GNN}_{\boldsymbol{\theta}_{\mathcal{S}}}(\mathbf{A}, \mathbf{X}), \mathbf{Y}\right) \quad \text{s.t} \quad \boldsymbol{\theta}_{\mathcal{S}} = \arg\min_{\boldsymbol{\theta}} \mathcal{L}(\text{GNN}_{\boldsymbol{\theta}}(\mathbf{A}', \mathbf{X}'), \mathbf{Y}'), \quad (1)$$

where $\text{GNN}_{\boldsymbol{\theta}}$ denotes the GNN model parameterized with $\boldsymbol{\theta}$, $\boldsymbol{\theta}_{\mathcal{S}}$ denotes the parameters of the model trained on $\mathcal{S}$, and $\mathcal{L}$ denotes the loss function used to measure the difference between model predictions and ground truth, i.e. cross-entropy loss. However, optimizing the above objective can lead to overfitting on a specific model initialization. To generate condensed data that generalizes to a distribution of random initializations $P_{\boldsymbol{\theta}_0}$, we rewrite the objective as follows:

$$\min_{\mathcal{S}} \text{E}_{\boldsymbol{\theta}_0 \sim P_{\boldsymbol{\theta}_0}} \left[\mathcal{L}\left(\text{GNN}_{\boldsymbol{\theta}_{\mathcal{S}}}(\mathbf{A}, \mathbf{X}), \mathbf{Y}\right)\right] \quad \text{s.t.} \quad \boldsymbol{\theta}_{\mathcal{S}} = \arg\min_{\boldsymbol{\theta}} \mathcal{L}(\text{GNN}_{\boldsymbol{\theta}(\boldsymbol{\theta}_0)}(\mathbf{A}', \mathbf{X}'), \mathbf{Y}'). \quad (2)$$

where $\boldsymbol{\theta}(\boldsymbol{\theta}_0)$ indicates that $\boldsymbol{\theta}$ is a function acting on $\boldsymbol{\theta}_0$. Note that the setting discussed above is for inductive learning where all the nodes are labeled and test nodes are unseen during training. We can easily generalize graph condensation to transductive setting by assuming $\mathbf{Y}$ is partially labeled.

### 3.1 GRAPH CONDENSATION VIA GRADIENT MATCHING

To tackle the optimization problem in Eq. (2), one strategy is to compute the gradient of $\mathcal{L}$ w.r.t $\mathcal{S}$ and optimize $\mathcal{S}$ via gradient descent, as in dataset distillation (Wang et al., 2018). However, this requires solving a nested loop optimization and unrolling the whole training trajectory of the inner problem, which can be prohibitively expensive. To bypass the bi-level optimization, we follow the gradient matching method proposed in (Zhao et al., 2021) which aims to match the network parameters w.r.t. large-real and small-synthetic training data by matching their gradients at each training step. In this way, the training trajectory on small-synthetic data $\mathcal{S}$ can mimic that on the large-real data $\mathcal{T}$, i.e., the models trained on these two datasets converge to similar solutions (parameters). Concretely, the parameter matching process for GNNs can be modeled as follows:

$$\min_{\mathcal{S}} \mathrm{E}_{\boldsymbol{\theta}_0 \sim P_{\boldsymbol{\theta}_0}} \left[ \sum_{t=0}^{T-1} D\left(\boldsymbol{\theta}_t^{\mathcal{S}}, \boldsymbol{\theta}_t^{\mathcal{T}}\right) \right] \quad \text{with}$$

$$\boldsymbol{\theta}_{t+1}^{\mathcal{S}} = \mathrm{opt}_{\boldsymbol{\theta}} \left( \mathcal{L}\left(\mathrm{GNN}_{\boldsymbol{\theta}_t^{\mathcal{S}}}(\mathbf{A}', \mathbf{X}'), \mathbf{Y}'\right) \right) \text{ and } \boldsymbol{\theta}_{t+1}^{\mathcal{T}} = \mathrm{opt}_{\boldsymbol{\theta}} \left( \mathcal{L}\left(\mathrm{GNN}_{\boldsymbol{\theta}_t^{\mathcal{T}}}(\mathbf{A}, \mathbf{X}), \mathbf{Y}\right) \right) \quad (3)$$

where $D(\cdot, \cdot)$ is a distance function, $T$ is the number of steps of the whole training trajectory, $\mathrm{opt}_{\boldsymbol{\theta}}$ is the update rule for model parameters, and $\boldsymbol{\theta}_t^{\mathcal{S}}, \boldsymbol{\theta}_t^{\mathcal{T}}$ denote the model parameters trained on $\mathcal{S}$ and $\mathcal{T}$ at time step $t$, respectively. Since our goal is to match the parameters step by step, we then consider one-step gradient descent for the update rule $\mathrm{opt}_{\boldsymbol{\theta}}$:

$$\boldsymbol{\theta}_{t+1}^{\mathcal{S}} \leftarrow \boldsymbol{\theta}_t^{\mathcal{S}} - \eta \nabla_{\boldsymbol{\theta}} \mathcal{L}\left(\mathrm{GNN}_{\boldsymbol{\theta}_t^{\mathcal{S}}}(\mathbf{A}', \mathbf{X}'), \mathbf{Y}'\right) \quad \text{and} \quad \boldsymbol{\theta}_{t+1}^{\mathcal{T}} \leftarrow \boldsymbol{\theta}_t^{\mathcal{T}} - \eta \nabla_{\boldsymbol{\theta}} \mathcal{L}\left(\mathrm{GNN}_{\boldsymbol{\theta}_t^{\mathcal{T}}}(\mathbf{A}, \mathbf{X}), \mathbf{Y}\right)$$
$$(4)$$

where $\eta$ is the learning rate for the gradient descent. Based on the observation made in Zhao et al. (2021) that the distance between $\boldsymbol{\theta}_t^{\mathcal{S}}$ and $\boldsymbol{\theta}_t^{\mathcal{T}}$ is typically small, we can simplify the objective as a gradient matching process as follows,

$$\min_{\mathcal{S}} \mathrm{E}_{\boldsymbol{\theta}_0 \sim P_{\theta_0}} \left[ \sum_{t=0}^{T-1} D\left(\nabla_{\boldsymbol{\theta}} \mathcal{L}\left(\mathrm{GNN}_{\boldsymbol{\theta}_t}(\mathbf{A}', \mathbf{X}'), \mathbf{Y}'\right), \nabla_{\boldsymbol{\theta}} \mathcal{L}\left(\mathrm{GNN}_{\boldsymbol{\theta}_t}(\mathbf{A}, \mathbf{X}), \mathbf{Y}\right)\right) \right] \quad (5)$$

where $\boldsymbol{\theta}_t^{\mathcal{S}}$ and $\boldsymbol{\theta}_t^{\mathcal{T}}$ are replaced by $\boldsymbol{\theta}_t$, which is trained on the small-synthetic graph. The distance $D$ is further defined as the sum of the distance $dis$ at each layer. Given two gradients $\mathbf{G}^{\mathcal{S}} \in \mathbb{R}^{d_1 \times d_2}$ and $\mathbf{G}^{\mathcal{T}} \in \mathbb{R}^{d_1 \times d_2}$ at a specific layer, the distance $dis(\cdot, \cdot)$ used for condensation is defined as follows,

$$dis(\mathbf{G}^{\mathcal{S}}, \mathbf{G}^{\mathcal{T}}) = \sum_{i=1}^{d_2} \left( 1 - \frac{\mathbf{G}_i^{\mathcal{S}} \cdot \mathbf{G}_i^{\mathcal{T}}}{\|\mathbf{G}_i^{\mathcal{S}}\| \|\mathbf{G}_i^{\mathcal{T}}\|} \right) \quad (6)$$

where $\mathbf{G}_i^{\mathcal{S}}, \mathbf{G}_i^{\mathcal{T}}$ are the $i$-th column vectors of the gradient matrices. With the above formulations, we are able to achieve parameter matching through an efficient strategy of gradient matching.

We note that jointly learning the three variables $\mathbf{A}', \mathbf{X}'$ and $\mathbf{Y}'$ is highly challenging, as they are interdependent. Hence, to simplify the problem, we fix the node labels $\mathbf{Y}'$ while keeping the class distribution the same as the original labels $\mathbf{Y}$.

**Graph Sampling.** GNNs are often trained in a full-batch manner (Kipf & Welling, 2017; Wu et al., 2019b). However, as suggested by previous works that reconstruct data from gradients (Zhu et al., 2019), large batch size tends to make reconstruction more difficult because more variables are involved during optimization. To make things worse, the computation cost of GNNs gets expensive on large graphs as the forward pass of GNNs involves the aggregation of enormous neighboring nodes. To address the above issues, we sample a fixed-size set of neighbors on the original graph in each aggregation layer of GNNs and adopt a mini-batch training strategy. To further reduce memory usage and ease optimization, we calculate the gradient matching loss for nodes from different classes separately, as matching the gradients w.r.t. the data from a single class is easier than that from all classes. Specifically, for a given class $c$, we sample a batch of nodes of class $c$ together with a portion of their neighbors from large-real data $\mathcal{T}$. We denote the process as $(\mathbf{A}_c, \mathbf{X}_c, \mathbf{Y}_c) \sim \mathcal{T}$. For the condensed graph $\mathbf{A}'$, we sample a batch of synthetic nodes of class $c$ but do not sample their neighbors. In other words, we use all of their neighbors, i.e., all other nodes, during the aggregation process, since we need to learn the connections with other nodes. We denote the process as $(\mathbf{A}'_c, \mathbf{X}'_c, \mathbf{Y}'_c) \sim \mathcal{S}$.

### 3.2 MODELING CONDENSED GRAPH DATA

One essential challenge in the graph condensation problem is how to model the condensed graph data and resolve dependency among nodes. The most straightforward way is to treat both $\mathbf{A}'$ and $\mathbf{X}'$

as free parameters. However, the number of parameters in $\mathbf{A}'$ grows quadratically as $N'$ increases. The increased model complexity can pose challenges in optimizing the framework and increase the risk of overfitting. Therefore, it is desired to parametrize the condensed adjacency matrix in a way where the number of parameters does not grow too fast. On the other hand, treating $\mathbf{A}'$ and $\mathbf{X}'$ as independent parameters overlooks the implicit correlations between graph structure and features, which have been widely acknowledged in the literature (III et al., 2014; Shalizi & Thomas, 2011); e.g., in social networks, users interact with others based on their interests, while in e-commerce, users purchase products due to certain product attributes. Hence, we propose to model the condensed graph structure as a function of the condensed node features:

$$\mathbf{A}' = g_\Phi(\mathbf{X}'), \qquad \text{with } \mathbf{A}'_{ij} = \text{Sigmoid}\left(\frac{\text{MLP}_\Phi([\mathbf{x}'_i; \mathbf{x}'_j]) + \text{MLP}_\Phi([\mathbf{x}'_j; \mathbf{x}'_i])}{2}\right) \qquad (7)$$

where $\text{MLP}_\Phi$ is a multi-layer neural network parameterized with $\Phi$ and $[\cdot; \cdot]$ denotes concatenation. In Eq. (7), we intentionally control $\mathbf{A}'_{ij} = \mathbf{A}'_{ji}$ to make the condensed graph structure symmetric since we are mostly dealing with symmetric graphs. It can also adjust to asymmetric graphs by setting $\mathbf{A}'_{ij} = \text{Sigmoid}(\text{MLP}_\Phi([\mathbf{x}_i; \mathbf{x}'_j]))$. Then we rewrite our objective as

$$\min_{\mathbf{X}', \Phi} \mathbb{E}_{\boldsymbol{\theta}_0 \sim P_{\theta_0}}\left[\sum_{t=0}^{T-1} D\left(\nabla_{\boldsymbol{\theta}} \mathcal{L}\left(\text{GNN}_{\boldsymbol{\theta}_t}(g_\Phi(\mathbf{X}'), \mathbf{X}'), \mathbf{Y}'\right), \nabla_{\boldsymbol{\theta}} \mathcal{L}\left(\text{GNN}_{\boldsymbol{\theta}_t}(\mathbf{A}, \mathbf{X}), \mathbf{Y}\right)\right)\right] \qquad (8)$$

Note that there are two clear benefits of the above formulation over the naïve one (free parameters). Firstly, the number of parameters for modeling graph structure no longer depends on the number of nodes, hence avoiding jointly learning $O(N'^2)$ parameters; as a result, when $N'$ gets larger, GCOND suffers less risk of overfitting. Secondly, if we want to grow the synthetic graph by adding more synthetic nodes condensed from real graph, the trained $\text{MLP}_\Phi$ can be employed to infer the connections of new synthetic nodes, and hence we only need to learn their features.

**Alternating Optimization Schema.** Jointly optimizing $\mathbf{X}'$ and $\Phi$ is often challenging as they are directly affecting each other. Instead, we propose to alternatively optimize $\mathbf{X}'$ and $\Phi$: we update $\Phi$ for the first $\tau_1$ epochs and then update $\mathbf{X}'$ for $\tau_2$ epochs; the process is repeated until the stopping condition is met – we find empirically that this does better as shown in Appendix C.

**Sparsification.** In the learned condensed adjacency matrix $\mathbf{A}'$, there can exist some small values which have little effect on the aggregation process in GNNs but still take up a certain amount of storage (e.g. 4 bytes per float). Thus, we remove the entries whose values are smaller than a given threshold $\delta$ to promote sparsity of the learned $\mathbf{A}'$. We further justify that suitable choices of $\delta$ for sparsification do not degrade performance a lot in Appendix C.

The detailed algorithm can be found in Algorithm 1 in Appendix B. In detail, we first set the condensed label set $\mathbf{Y}'$ to fixed values and initialize $\mathbf{X}'$ as node features randomly selected from each class. In each outer loop, we sample a GNN model initialization $\boldsymbol{\theta}$ from a distribution $P_{\boldsymbol{\theta}}$. Then, for each class we sample the corresponding node batches from $\mathcal{T}$ and $\mathcal{S}$, and calculate the gradient matching loss within each class. The sum of losses from different classes are used to update $\mathbf{X}'$ or $\Phi$. After that we update the GNN parameters for $\tau_{\boldsymbol{\theta}}$ epochs. When finishing the updating of condensed graph parameters, we use $\mathbf{A}' = \text{ReLU}(g_\Phi(\mathbf{X}') - \delta)$ to obtain the final sparsified graph structure.

**A "Graphless" Model Variant.** We now explore another parameterization for the condensed graph data. We provide a model variant named GCOND-X that only learns the condensed node features $\mathbf{X}'$ without learning the condensed structure $\mathbf{A}'$. In other words, we use a fixed identity matrix $\mathbf{I}$ as the condensed graph structure. Specifically, this model variant aims to match the gradients of GNN parameters on the large-real data $(\mathbf{A}, \mathbf{X})$ and small-synthetic data $(\mathbf{I}, \mathbf{X}')$. Although GCOND-X is unable to learn the condensed graph structure which can be highly useful for downstream data analysis, it still shows competitive performance in Table 2 in the experiments because the features are learned to incorporate relevant information from the graph via the matching loss.

## 4 EXPERIMENTS

In this section, we design experiments to validate the effectiveness of the proposed framework GCOND. We first introduce experimental settings, then compare GCOND against representative baselines with discussions and finally show some advantages of GCOND.

## 4.1 EXPERIMENTAL SETUP

**Datasets.** We evaluate the condensation performance of the proposed framework on three transductive datasets, i.e., Cora, Citeseer (Kipf & Welling, 2017) and Ogbn-arxiv (Hu et al., 2020), and two inductive datasets, i.e., Flickr (Zeng et al., 2020) and Reddit (Hamilton et al., 2017). We use the public splits for all the datasets. For the inductive setting, we follow the setup in (Hamilton et al., 2017) where the test graph is not available during training. Dataset statistics are shown in Appendix A.

**Baselines.** We compare our proposed methods to five baselines: (i) one *graph coarsening* method (Loukas, 2019; Huang et al., 2021b), (ii-iv) three coreset methods (*Random*, *Herding* (Welling, 2009) and *K-Center* (Farahani & Hekmatfar, 2009; Sener & Savarese, 2018)), and (v) *dataset condensation* (DC). For the graph coarsening method, we adopt the variation neighborhoods method implemented by Huang et al. (2021b). For coreset methods, we first use them to select nodes from the original dataset and induce a subgraph from the selected nodes to serve as the reduced graph. In Random, the nodes are randomly selected. The Herding method, which is often used in continual learning (Rebuffi et al., 2017; Castro et al., 2018), picks samples that are closest to the cluster center. K-Center selects the center samples to minimize the largest distance between a sample and its nearest center. We use the implementations provided by Zhao et al. (2021) for Herding, K-Center and DC. As vanilla DC cannot leverage any structure information, we develop a variant named DC-Graph, which additionally leverages graph structure during test stage, to replace DC for the following experiments. A comparison between DC, DC-Graph, GCOND and GCOND-X is shown in Table 1 and their training details can be found in Appendix A.3.

**Evaluation.** We first use the aforementioned baselines to obtain condensed graphs and then evaluate them on GNNs for both transductive and inductive node classification tasks. For transductive datasets, we condense the full graph with $N$ nodes into a synthetic graph with $rN$ ($0 < r < 1$) nodes, where $r$ is the ratio of synthetic nodes to original nodes. For inductive datasets, we only condense the training graph since the rest of the full graph is not available during training. The choices of $r$[2] are listed in Table 2. For each $r$, we generate 5 condensed graphs with different seeds. To evaluate the effectiveness of condensed graphs, we have two stages: (1) a training stage, where we train a GNN model on the condensed graph, and (2) a test stage, where the trained GNN uses the test graph (or full graph in transductive setting) to infer the labels for test nodes. The resulting test performance is compared with that obtained when training on original datasets. All experiments are repeated 10 times, and we report average performance and variance.

**Hyperparameter settings.** As our goal is to generate highly informative synthetic graphs which can benefit GNNs, we choose one representative model, GCN (Kipf & Welling, 2017), for performance evaluation. For the GNN used in condensation, i.e., the $\text{GNN}_\theta(\cdot)$ in Eq. (8), we adopt SGC (Wu et al., 2019a) which decouples the propagation and transformation process but still shares similar graph filtering behavior as GCN. Unless otherwise stated, we use 2-layer models with 256 hidden units. The weight decay and dropout for the models are set to 0 in condensation process. More details for hyper-parameter tuning can be found in Appendix A.

## 4.2 COMPARISON WITH BASELINES

In this subsection, we test the performance of a 2-layer GCN on the condensed graphs, and compare the proposed GCOND and GCOND-X with baselines. Notably, all methods produce both structure and node features, i.e. $\mathbf{A}'$ and $\mathbf{X}'$, except DC-Graph and GCOND-X. Since DC-Graph and GCOND-X do not produce any structure, we simply use an identity matrix as the adjacency matrix when training GNNs solely on condensed features. However, during inference, we use the full graph (transductive setting) or test graph (inductive setting) to propagate information based on the trained GNNs. This training paradigm is similar to the C&S model (Huang et al., 2021a) which trains an MLP without the graph information and performs label propagation based on MLP predictions. Table 2 reports node classification performance; we make the following observations:

**Obs 1. Condensation methods achieve promising performance even with extremely large reduction rates.** Condensation methods, i.e., GCOND, GCOND-X and DC-Graph, outperform coreset methods and graph coarsening significantly at the lowest ratio $r$ for each dataset. This shows the importance of learning synthetic data using the guidance from downstream tasks. Notably, GCOND

---

[2]We determine $r$ based on original graph size and labeling rate – see Appendix A for details.

Table 1: Information comparison used during condensation, training and test for reduction methods. $\mathbf{A}', \mathbf{X}'$ and $\mathbf{A}, \mathbf{X}$ are condensed (original) graph and features, respectively.

| | DC | DC-Graph | GCOND-X | GCOND |
|---|---|---|---|---|
| Condensation | $\mathbf{X}_{\text{train}}$ | $\mathbf{X}_{\text{train}}$ | $\mathbf{A}_{\text{train}}, \mathbf{X}_{\text{train}}$ | $\mathbf{A}_{\text{train}}, \mathbf{X}_{\text{train}}$ |
| Training | $\mathbf{X}'$ | $\mathbf{X}'$ | $\mathbf{X}'$ | $\mathbf{A}', \mathbf{X}'$ |
| Test | $\mathbf{X}_{\text{test}}$ | $\mathbf{A}_{\text{test}}, \mathbf{X}_{\text{test}}$ | $\mathbf{A}_{\text{test}}, \mathbf{X}_{\text{test}}$ | $\mathbf{A}_{\text{test}}, \mathbf{X}_{\text{test}}$ |

Table 2: GCOND and GCOND-X achieves promising performance in comparison to baselines even with extremely large reduction rates. We report transductive performance on Citeseer, Cora, Ogbn-arxiv; inductive performance on Flickr, Reddit. Performance is reported as test accuracy (%).

| | | | | Baselines | | | Proposed | | |
|---|---|---|---|---|---|---|---|---|---|
| Dataset | Ratio $(r)$ | Random $(\mathbf{A}', \mathbf{X}')$ | Herding $(\mathbf{A}', \mathbf{X}')$ | K-Center $(\mathbf{A}', \mathbf{X}')$ | Coarsening $(\mathbf{A}', \mathbf{X}')$ | DC-Graph $(\mathbf{X}')$ | GCOND-X $(\mathbf{X}')$ | GCOND $(\mathbf{A}', \mathbf{X}')$ | Whole Dataset |
|---|---|---|---|---|---|---|---|---|---|
| Citeseer | 0.9% | 54.4±4.4 | 57.1±1.5 | 52.4±2.8 | 52.2±0.4 | 66.8±1.5 | **71.4±0.8** | 70.5±1.2 | |
| | 1.8% | 64.2±1.7 | 66.7±1.0 | 64.3±1.0 | 59.0±0.5 | 66.9±0.9 | 69.8±1.1 | **70.6±0.9** | 71.7±0.1 |
| | 3.6% | 69.1±0.1 | 69.0±0.1 | 69.1±0.1 | 65.3±0.5 | 66.3±1.5 | 69.4±1.4 | **69.8±1.4** | |
| Cora | 1.3% | 63.6±3.7 | 67.0±1.3 | 64.0±2.3 | 31.2±0.2 | 67.3±1.9 | 75.9±1.2 | **79.8±1.3** | |
| | 2.6% | 72.8±1.1 | 73.4±1.0 | 73.2±1.2 | 65.2±0.6 | 67.6±3.5 | 75.7±0.9 | **80.1±0.6** | 81.2±0.2 |
| | 5.2% | 76.8±0.1 | 76.8±0.1 | 76.7±0.1 | 70.6±0.1 | 67.7±2.2 | 76.0±0.9 | **79.3±0.3** | |
| Ogbn-arxiv | 0.05% | 47.1±3.9 | 52.4±1.8 | 47.2±3.0 | 35.4±0.3 | 58.6±0.4 | **61.3±0.5** | 59.2±1.1 | |
| | 0.25% | 57.3±1.1 | 58.6±1.2 | 56.8±0.8 | 43.5±0.2 | 59.9±0.3 | **64.2±0.4** | 63.2±0.3 | 71.4±0.1 |
| | 0.5% | 60.0±0.9 | 60.4±0.8 | 60.3±0.4 | 50.4±0.1 | 59.5±0.3 | 63.1±0.5 | **64.0±0.4** | |
| Flickr | 0.1% | 41.8±2.0 | 42.5±1.8 | 42.0±0.7 | 41.9±0.2 | 46.3±0.2 | 45.9±0.1 | **46.5±0.4** | |
| | 0.5% | 44.0±0.4 | 43.9±0.9 | 43.2±0.1 | 44.5±0.1 | 45.9±0.1 | 45.0±0.2 | **47.1±0.1** | 47.2±0.1 |
| | 1% | 44.6±0.2 | 44.4±0.6 | 44.1±0.4 | 44.6±0.1 | 45.8±0.1 | 45.0±0.1 | **47.1±0.1** | |
| Reddit | 0.05% | 46.1±4.4 | 53.1±2.5 | 46.6±2.3 | 40.9±0.5 | 88.2±0.2 | **88.4±0.4** | 88.0±1.8 | |
| | 0.1% | 58.0±2.2 | 62.7±1.0 | 53.0±3.3 | 42.8±0.8 | 89.5±0.1 | 89.3±0.1 | **89.6±0.7** | 93.9±0.0 |
| | 0.2% | 66.3±1.9 | 71.0±1.6 | 58.5±2.1 | 47.4±0.9 | **90.5±1.2** | 88.8±0.4 | 90.1±0.5 | |

achieves 79.8%, 80.1% and 79.3% at 1.3%, 2.6% and 5.2% condensation ratios at Cora, while the whole dataset performance is 81.2%. The GCOND variants also show promising performance on Cora, Flickr and Reddit at all coarsening ratios. Although the gap between whole-dataset Ogbn-arxiv and our methods is larger, they still outperform baselines by a large margin.

**Obs 2. Learning $\mathbf{X}'$ instead of $(\mathbf{A}', \mathbf{X}')$ as the condensed graph can also lead to good results.** GCOND-X achieves close performance to GCOND on 11 of 15 cases. Since our objective in graph condensation is to achieve parameter matching through gradient matching, training a GNN on the learned features $\mathbf{X}'$ with identity adjacency matrix is also able to mimic the training trajectory of GNN parameters. One reason could be that $\mathbf{X}'$ has already encoded node features and structural information of the original graph during the condensation process. However, there are many scenarios where the graph structure is essential such as the generalization to other GNN architectures (e.g., GAT) and visualizing the patterns in the data. More details are given in the following subsections.

**Obs 3. Condensing node features and structural information simultaneously can lead to better performance.** In most cases, GCOND and GCOND-X obtain much better performance than DC-Graph. One key reason is that GCOND and GCOND-X can take advantage of both node features and structural information in the condensation process. We notice that DC-Graph achieves a highly comparable result (90.5%) on Reddit at 0.2% condensation ratio to the whole dataset performance (93.9%). This may indicate that the original training graph structure might not be useful. To verify this assumption, we train a GCN on the original Reddit dataset without using graph structure (i.e., setting $\mathbf{A}_{\text{train}} = \mathbf{I}$), but allow using the test graph structure for inference using the trained model. The obtained performance is 92.5%, which is very close to the original performance 93.9%, indicating that training without graph structure can still achieve comparable performance. We also note that learning $\mathbf{X}', \mathbf{A}'$ simultaneously creates opportunities to absorb information from graph structure directly into learned features, lessening reliance on distilling graph properties reliably while still achieving good generalization performance from features.

**Obs 4. Larger condensed graph size does not strictly indicate better performance.** Although larger condensed graph sizes allow for more parameters which can potentially encapsulate more information from original graph, it simultaneously becomes harder to optimize due to the increased

Table 3: Graph condensation can work well with different architectures. Avg. stands for the average test accuracy of APPNP, Cheby, GCN, GraphSAGE and SGC. SAGE stands for GraphSAGE.

| | Methods | Data | MLP | GAT | APPNP | Cheby | GCN | SAGE | SGC | Avg. |
|---|---|---|---|---|---|---|---|---|---|---|
| Citeseer $r = 1.8\%$ | DC-Graph | $\mathbf{X}'$ | 66.2 | - | 66.4 | 64.9 | 66.2 | 65.9 | 69.6 | 66.6 |
| | GCOND-X | $\mathbf{X}'$ | 69.6 | - | 69.7 | 70.6 | 69.7 | 69.2 | 71.6 | 70.2 |
| | GCOND | $\mathbf{A}', \mathbf{X}'$ | 63.9 | 55.4 | 69.6 | 68.3 | 70.5 | 66.2 | 70.3 | 69.0 |
| Cora $r = 2.6\%$ | DC-Graph | $\mathbf{X}'$ | 67.2 | - | 67.1 | 67.7 | 67.9 | 66.2 | 72.8 | 68.3 |
| | GCOND-X | $\mathbf{X}'$ | 76.0 | - | 77.0 | 74.1 | 75.3 | 76.0 | 76.1 | 75.7 |
| | GCOND | $\mathbf{A}', \mathbf{X}'$ | 73.1 | 66.2 | 78.5 | 76.0 | 80.1 | 78.2 | 79.3 | 78.4 |
| Ogbn-arxiv $r = 0.25\%$ | DC-Graph | $\mathbf{X}'$ | 59.9 | - | 60.0 | 55.7 | 59.8 | 60.0 | 60.4 | 59.2 |
| | GCOND-X | $\mathbf{X}'$ | 64.1 | - | 61.5 | 59.5 | 64.2 | 64.4 | 64.7 | 62.9 |
| | GCOND | $\mathbf{A}', \mathbf{X}'$ | 62.2 | 60.0 | 63.4 | 54.9 | 63.2 | 62.6 | 63.7 | 61.6 |
| Flickr $r = 0.5\%$ | DC-Graph | $\mathbf{X}'$ | 43.1 | - | 45.7 | 43.8 | 45.9 | 45.8 | 45.6 | 45.4 |
| | GCOND-X | $\mathbf{X}'$ | 42.1 | - | 44.6 | 42.3 | 45.0 | 44.7 | 44.4 | 44.2 |
| | GCOND | $\mathbf{A}', \mathbf{X}'$ | 44.8 | 40.1 | 45.9 | 42.8 | 47.1 | 46.2 | 46.1 | 45.6 |
| Reddit $r = 0.1\%$ | DC-Graph | $\mathbf{X}'$ | 50.3 | - | 81.2 | 77.5 | 89.5 | 89.7 | 90.5 | 85.7 |
| | GCOND-X | $\mathbf{X}'$ | 40.1 | - | 78.7 | 74.0 | 89.3 | 89.3 | 91.0 | 84.5 |
| | GCOND | $\mathbf{A}', \mathbf{X}'$ | 42.5 | 60.2 | 87.8 | 75.5 | 89.4 | 89.1 | 89.6 | 86.3 |

model complexity. We observe that once the condensation ratio reaches a certain threshold, the performance becomes stable. However, the performance of coreset methods and graph coarsening is much more sensitive to the reduction ratio. Coreset methods only select existing samples while graph coarsening groups existing nodes into super nodes. When the reduction ratio is too low, it becomes extremely difficult to select informative nodes or form representative super nodes by grouping.

## 4.3 GENERALIZABILITY OF CONDENSED GRAPHS

Next, we illustrate the generalizability of condensed graphs from the following three perspectives.

**Different Architectures.** Next, we show the generalizability of the graph condensation procedure. Specifically, we show test performance when using a graph condensed by one GNN model to train different GNN architectures. Specifically, we choose APPNP (Klicpera et al., 2019), GCN, SGC (Wu et al., 2019a), GraphSAGE (Hamilton et al., 2017), Cheby (Defferrard et al., 2016) and GAT (Velickovic et al., 2018). We also include MLP and report the results in Table 3. From the table, we find that the condensed graphs generated by GCOND show good generalization on different architectures. We may attribute such transferability across different architectures to similar filtering behaviors of those GNN models, which have been studied in Ma et al. (2020); Zhu et al. (2021).

**Versatility of GCOND.** The proposed GCOND is highly composable in that we can adopt various GNNs inside the condensation network. We investigate the performances of various GNNs when using different GNN models in the condensation process, i.e., $\text{GNN}_\theta(\cdot)$ in Eq. (8). We choose APPNP, Cheby, GCN, GraphSAGE and SGC to serve as the models used in condensation and evaluation. Note that we omit GAT due to its deterioration under large neighborhood sizes (Ma et al., 2021). We choose Cora and Ogbn-arxiv to report the performance in Table 4 where C and T denote condensation and test models, respectively. The graphs condensed by different GNNs all show strong transfer performance on other architectures.

**Neural Architecture Search.** We also perform experiments on neural architecture search, detailed in Appendix C.2. We search 480 architectures of APPNP and perform the search process on Cora, Citeseer and Ogbn-arxiv. Specifically, we train each architecture on the reduced graph for epochs on as the model converges faster on the smaller graph. We observe reliable correlation of performances between condensed dataset training and whole-dataset training as shown in Table 9: 0.76/0.79/0.64 for Cora/Citeseer/Ogbn-arxiv.

## 4.4 ANALYSIS ON CONDENSED DATA

**Statistics of Condensed Graphs.** In Table 5, we compare several properties between condensed graphs and original graphs. Note that a widely used homophily measure is defined in (Zhu et al., 2020) but it does not apply to weighted graphs. Hence, when computing homophily, we binarize the graphs by removing edges whose weights are smaller than 0.5. We make the following observations. First, while achieving similar performance for downstream tasks, the condensed graphs contain fewer nodes and take much less storage. Second, the condensed graphs are less sparse than their

Table 4: Cross-architecture performance is shown in test accuracy (%). SAGE: GraphSAGE. Graphs condensed by different GNNs all show strong transfer performance on other architectures.

(a) Cora, $r$=2.6%

| C\T | APPNP | Cheby | GCN | SAGE | SGC |
|---|---|---|---|---|---|
| APPNP | 72.1±2.6 | 60.8±6.4 | **73.5**±2.4 | 72.3±3.5 | 73.1±3.1 |
| Cheby | 75.3±2.9 | 71.8±1.1 | **76.8**±2.1 | 76.4±2.0 | 75.5±3.5 |
| GCN | 69.8±4.0 | 53.2±3.4 | **70.6**±3.7 | 60.2±1.9 | 68.7±5.4 |
| SAGE | 77.1±1.1 | 69.3±1.7 | 77.0±0.7 | 76.1±0.7 | **77.7**±1.8 |
| SGC | 78.5±1.0 | 76.0±1.1 | **80.1**±0.6 | 78.2±0.9 | 79.3±0.7 |

(b) Ogbn-arxiv, $r$=0.05%

| C\T | APPNP | Cheby | GCN | SAGE | SGC |
|---|---|---|---|---|---|
| APPNP | 60.3±0.2 | 51.8±0.5 | 59.9±0.4 | 59.0±1.1 | **61.2**±0.4 |
| Cheby | 57.4±0.4 | 53.5±0.5 | 57.4±0.8 | 57.1±0.8 | **58.2**±0.6 |
| GCN | 59.3±0.4 | 51.8±0.7 | **60.3**±0.3 | 60.2±0.4 | 59.2±0.7 |
| SAGE | 57.6±0.8 | 53.9±0.6 | 58.1±0.6 | 57.8±0.7 | **59.0**±1.1 |
| SGC | 59.7±0.5 | 49.5±0.8 | 59.2±1.1 | 58.9±1.6 | **60.5**±0.6 |

Table 5: Comparison between condensed graphs and original graphs. The condensed graphs have fewer nodes and are more dense.

| | Citeseer, $r$=0.9% | | Cora, $r$=1.3% | | Ogbn-arxiv, $r$=0.25% | | Flickr, $r$=0.5% | | Reddit, $r$=0.1% | |
|---|---|---|---|---|---|---|---|---|---|---|
| | Whole | GCOND | Whole | GCOND | Whole | GCOND | Whole | GCOND | Whole | GCOND |
| Accuracy | 70.7 | 70.5 | 81.5 | 79.8 | 71.4 | 63.2 | 47.1 | 47.1 | 94.1 | 89.4 |
| #Nodes | 3,327 | 60 | 2,708 | 70 | 169,343 | 454 | 44,625 | 223 | 153,932 | 153 |
| #Edges | 4,732 | 1,454 | 5,429 | 2,128 | 1,166,243 | 3,354 | 218,140 | 3,788 | 10,753,238 | 301 |
| Sparsity | 0.09% | 80.78% | 0.15% | 86.86% | 0.01% | 3.25% | 0.02% | 15.23% | 0.09% | 2.57% |
| Homophily | 0.74 | 0.65 | 0.81 | 0.79 | 0.65 | 0.07 | 0.33 | 0.28 | 0.78 | 0.04 |
| Storage | 47.1 MB | 0.9 MB | 14.9 MB | 0.4 MB | 100.4 MB | 0.3 MB | 86.8 MB | 0.5 MB | 435.5 MB | 0.4 MB |

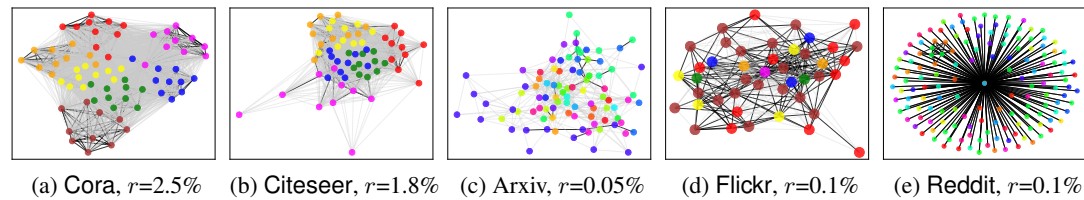

(a) Cora, $r$=2.5%  (b) Citeseer, $r$=1.8%  (c) Arxiv, $r$=0.05%  (d) Flickr, $r$=0.1%  (e) Reddit, $r$=0.1%

Figure 2: Condensed graphs sometimes exhibit structure mimicking the original (a, b, d). Other times (c, e), learned features absorb graph properties and create less explicit graph reliance.

larger counterparts. Since the condensed graph is on extremely small scale, there would be almost no connections between nodes if the condensed graph maintains the original sparsity. Third, for Citeseer, Cora and Flickr, the homophily information are well preserved in the condensed graphs.

**Visualization.** We present the visualization results for all datasets in Figure 4, where nodes with the same color are from the same class. Notably, as the learned condensed graphs are weighted graphs, we use black lines to denote the edges with weights larger than 0.5 and gray lines to denote the edges with weights smaller than 0.5. From Figure 4, we can observe some patterns in the condensed graphs, e.g., the homophily patterns on Cora and Citeseer are well preserved. Interestingly, the learned graph for Reddit is very close to a star graph where almost all the nodes only have connections with very few center nodes. Such a structure can be meaningless for GNNs because almost all the nodes receive the information from their neighbors. In this case, the learned features $\mathbf{X}'$ play a major role in training GNN parameters, indicating that the original training graph of Reddit is not very informative, aligning with our observations in Section 4.2.

## 5 CONCLUSION

The prevalence of large-scale graphs poses great challenges in training graph neural networks. Thus, we study a novel problem of *graph condensation* which targets at condensing a large-real graph into a small-synthetic one while maintaining the performances of GNNs. Through our proposed framework, we are able to significantly reduce the graph size while approximating the original performance. The condensed graphs take much less space of storage and can be used to efficiently train various GNN architectures. Future work can be done on (1) improving the transferability of condensed graphs for different GNNs, (2) studying graph condensation for other tasks such as graph classification and (3) designing condensation framework for multi-label datasets.

ACKNOLWEDGEMENT

Wei Jin and Jiliang Tang are supported by the National Science Foundation (NSF) under grant numbers IIS1714741, CNS1815636, IIS1845081, IIS1907704, IIS1928278, IIS1955285, IOS2107215, and IOS2035472, the Army Research Office (ARO) under grant number W911NF-21-1-0198, the Home Depot, Cisco Systems Inc. and SNAP Inc.

ETHICS STATEMENT

To the best of our knowledge, there are no ethical issues with this paper.

REPRODUCIBILITY STATEMENT

To ensure reproducibility of our experiments, we provide our source code at `https://github.com/ChandlerBang/GCond`. The hyper-parameters are described in details in the appendix. We also provide a pseudo-code implementation of our framework in the appendix.

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

# A    DATASETS AND HYPER-PARAMETERS

## A.1    DATASETS

We evaluate the proposed framework on three transductive datasets, i.e., Cora, Citeseer (Kipf & Welling, 2017) and Ogbn-arxiv (Hu et al., 2020), and two inductive datasets, i.e., Flickr (Zeng et al., 2020) and Reddit (Hamilton et al., 2017). Since all the datasets have public splits, we download them from PyTorch Geometric (Fey & Lenssen, 2019) and use those splits throughout the experiments. Dataset statistics are shown in Table 6.

Table 6: Dataset statistics. The first three are transductive datasets and the last two are inductive datasets.

| Dataset | #Nodes | #Edges | #Classes | #Features | Training/Validation/Test |
|---|---|---|---|---|---|
| Cora | 2,708 | 5,429 | 7 | 1,433 | 140/500/1000 |
| Citeseer | 3,327 | 4,732 | 6 | 3,703 | 120/500/1000 |
| Ogbn-arxiv | 169,343 | 1,166,243 | 40 | 128 | 90,941/29,799/48,603 |
| Flickr | 89,250 | 899,756 | 7 | 500 | 44,625/22312/22313 |
| Reddit | 232,965 | 57,307,946 | 210 | 602 | 15,3932/23,699/55,334 |

## A.2    HYPER-PARAMETER SETTING

**Condensation Process.** For DC, we tune the number of hidden layers in a range of $\{1, 2, 3\}$ and fix the number of hidden units to 256. We further tune the number of epochs for training DC in a range of $\{100, 200, 400\}$. For GCOND, without specific mention, we adopt a 2-layer SGC (Wu et al., 2019a) with 256 hidden units as the GNN used for gradient matching. The function $g_\Phi$ that models the relationship between $\mathbf{A}'$ and $\mathbf{X}'$ is implemented as a multi-layer perceptron (MLP). Specifically, we adopt a 3-layer MLP with 128 hidden units for small graphs (Cora and Citeseer) and 256 hidden units for large graphs (Flickr, Reddit and Ogbn-arxiv). We tune the training epoch for GCOND in a range of $\{400, 600, 1000\}$. For GCOND-X, we tune the number of hidden layers in a range of $\{1, 2, 3\}$ and fix the number of hidden units to 256. We further tune the number of epochs for training GCOND-X in a range of $\{100, 200, 400\}$. We tune the learning rate for all the methods in a range of $\{0.1, 0.01, 0.001, 0.0001\}$. Furthermore, we set $\delta$ to be $0.05, 0.05, 0.01, 0.01, 0.01$ for Citeseer, Cora, Ogbn-arxiv, Flickr and Reddit, respectively.

For the choices of condensation ratio $r$, we divide the discussion into two parts. The first part is about transductive datasets. For Cora and Citeseer, since their labeling rates are very small (5.2% and 3.6%, respectively), we choose $r$ to be $\{25\%, 50\%, 100\%\}$ of the labeling rate. Thus, we finally choose $\{1.3\%, 2.6\%, 5.2\%\}$ for Cora and $\{0.9\%, 1.8\%, 3.6\%\}$ for Citeseer. For Ogbn-arxiv, we choose $r$ to be $\{0.1\%, 0.5\%, 1\%\}$ of its labeling rate (53%), thus being $\{0.05\%, 0.25\%, 0.5\%\}$. The second part is about inductive datasets. As the nodes in the training graphs are all labeled in inductive datasets, we simply choose $\{0.1\%, 0.5\%, 0.1\%\}$ for Flickr and $0.05\%, 0.1\%, 0.2\%$ for Reddit.

**Evaluation Process.** During the evaluation process, we set dropout rate to be 0 and weight decay to be 0.0005 when training various GNNs. The number of epochs is set to 3000 for GAT while it is set to 600 for other models. The initial learning rate is set to 0.01.

**Settings for Table 3 and Table 4.** In both condensation stage and evaluation stage, we set the depth of GNNs to 2. During condensation stage, we set weight decay to 0, dropout to 0 and training epochs to 1000. During evaluation stage, we set weight decay to 0.0005, dropout to 0 and training epochs to 600.

## A.3    TRAINING DETAILS OF DC-GRAPH, GCOND-X AND GCOND

**DC-Graph**: During the condensation stage, DC-Graph only leverages the node features to produce condensed node features $\mathbf{X}'$. During the training stage of evaluation, DC-Graph takes the condensed features $\mathbf{X}'$ together with an identity matrix as the graph structure to train a GNN. In the later test stage of evaluation, the GNN takes both test node features and test graph structure as input to make predictions for test nodes.

**GCOND-X**: During the condensation stage, GCOND-X leverages both the graph structure and node features to produce condensed node features $\mathbf{X}'$. During the training stage of evaluation, GCOND-X takes the condensed features $\mathbf{X}'$ together with an identity matrix as the graph structure to train a GNN. In the later test stage of evaluation, the GNN takes both test node features and test graph structure as input to make predictions for test nodes.

**GCOND**: During the condensation stage, GCOND leverages both the graph structure and node features to produce condensed graph data $(\mathbf{A}', \mathbf{X}')$. During the training stage of evaluation, GCOND takes the condensed data $(\mathbf{A}', \mathbf{X}')$ to train a GNN. In the later test stage of evaluation, the GNN takes both test node features and test graph structure as input to make predictions for test nodes.

## B  ALGORITHM

We show the detailed algorithm of GCOND in Algorithm 1. In detail, we first set the condensed label set $\mathbf{Y}'$ to fixed values and initialize $\mathbf{X}'$ as node features randomly selected from each class. In each outer loop, we sample a GNN model initialization $\boldsymbol{\theta}$ from a distribution $P_{\boldsymbol{\theta}}$. Then, for each class we sample the corresponding node batches from $\mathcal{T}$ and $\mathcal{S}$, and calculate the gradient matching loss within each class. The sum of losses from different classes are used to update $\mathbf{X}'$ or $\Phi$. After that we update the GNN parameters for $\tau_{\boldsymbol{\theta}}$ epochs. When finishing the updating of condensed graph parameters, we use $\mathbf{A}' = \text{ReLU}(g_\Phi(\mathbf{X}') - \delta)$ to obtain the final sparsified graph structure.

---

**Algorithm 1:** GCOND for Graph Condensation

---

1 **Input:** Training data $\mathcal{T} = (\mathbf{A}, \mathbf{X}, \mathbf{Y})$, pre-defined condensed labels $\mathbf{Y}'$
2 Initialize $\mathbf{X}'$ as node features randomly selected from each class
3 **for** $k = 0, \ldots, K - 1$ **do**
4 $\quad$ Initialize $\boldsymbol{\theta}_0 \sim P_{\boldsymbol{\theta}_0}$
5 $\quad$ **for** $t = 0, \ldots, T - 1$ **do**
6 $\quad\quad$ $D' = 0$
7 $\quad\quad$ **for** $c = 0, \ldots, C - 1$ **do**
8 $\quad\quad\quad$ Compute $\mathbf{A}' = g_\Phi(\mathbf{X}')$; then $\mathcal{S} = \{\mathbf{A}', \mathbf{X}', \mathbf{Y}'\}$
9 $\quad\quad\quad$ Sample $(\mathbf{A}_c, \mathbf{X}_c, \mathbf{Y}_c) \sim \mathcal{T}$ and $(\mathbf{A}'_c, \mathbf{X}'_c, \mathbf{Y}'_c) \sim \mathcal{S}$ $\qquad \triangleright$ detailed in Section 3.1
10 $\quad\quad\quad$ Compute $\mathcal{L}^{\mathcal{T}} = \mathcal{L}\left(\text{GNN}_{\boldsymbol{\theta}_t}(\mathbf{A}_c, \mathbf{X}_c), \mathbf{Y}_c\right)$ and $\mathcal{L}^{\mathcal{S}} = \mathcal{L}\left(\text{GNN}_{\boldsymbol{\theta}_t}(\mathbf{A}'_c, \mathbf{X}'_c), \mathbf{Y}'_c\right)$
11 $\quad\quad\quad$ $D' \leftarrow D' + D(\nabla_{\boldsymbol{\theta}_t}\mathcal{L}^{\mathcal{T}}, \nabla_{\boldsymbol{\theta}_t}\mathcal{L}^{\mathcal{S}})$
12 $\quad\quad$ **if** $t\%(\tau_1 + \tau_2) < \tau_1$ **then**
13 $\quad\quad\quad$ Update $\mathbf{X}' \leftarrow \mathbf{X}' - \eta_1 \nabla_{\mathbf{X}'} D'$
14 $\quad\quad$ **else**
15 $\quad\quad\quad$ Update $\Phi \leftarrow \Phi - \eta_2 \nabla_\Phi D'$
16 $\quad\quad$ Update $\boldsymbol{\theta}_{t+1} \leftarrow \text{opt}_{\boldsymbol{\theta}}(\boldsymbol{\theta}_t, \mathcal{S}, \tau_{\boldsymbol{\theta}})$ $\qquad \triangleright \tau_{\boldsymbol{\theta}}$ is the number of steps for updating $\boldsymbol{\theta}$
17 $\mathbf{A}' = \text{ReLU}(g_\Phi(\mathbf{X}') - \delta)$
18 **Return:** $(\mathbf{A}', \mathbf{X}', \mathbf{Y}')$

---

## C  MORE EXPERIMENTS

### C.1  ABLATION STUDY

**Different Parameterization.** We study the effect of different parameterizations for modeling $\mathbf{A}'$ and compare GCOND with modeling $\mathbf{A}'$ as free parameters in Table 7. From the table, we observe a significant improvement by taking into account the relationship between $\mathbf{A}'$ and $\mathbf{X}'$. This suggests that directly modeling the structure as a function of features can ease the optimization and lead to better condensed graph data.

**Joint optimization versus alternate optimization.** We perform the ablation study on joint optimization and alternate optimization when updating $\Phi$ and $\mathbf{X}'$. The results are shown in Table 8. From the table, we can observe that joint optimization always gives worse performance and the standard deviation is much higher than alternate optimization.

Table 7: Ablation study on different parametrizations.

| Parameters | Citeseer, $r=1.8\%$ | Cora, $r=2.6\%$ | Ogbn-arxiv, $r=0.25\%$ |
|---|---|---|---|
| $\mathbf{A}', \mathbf{X}'$ | 62.2±4.8 | 75.5±0.6 | 63.0±0.5 |
| $\Phi, \mathbf{X}'$ | 70.6±0.9 | 80.1±0.6 | 63.2±0.3 |

Table 8: Ablation study on different optimization strategies.

| | Citeseer, $r=1.8\%$ | Cora, $r=2.6\%$ | Ogbn-arxiv, $r=0.25\%$ | Flickr, $r=0.5\%$ | Reddit, $r=0.1\%$ |
|---|---|---|---|---|---|
| Joint | 68.2±3.0 | 79.9±1.6 | 62.8±1.1 | 45.4±0.4 | 87.5±1.8 |
| Alternate | 70.6±0.9 | 80.1±0.6 | 63.2±0.3 | 47.1±0.1 | 89.5±0.8 |

## C.2 NEURAL ARCHITECTURE SEARCH

We focus on APPNP instead of GCN since the architecture of APPNP involves more hyper-parameters regarding its architecture setup. The detailed search space is shown as follows:

(a) **Number of propagation $K$:** we search the number of propagation $K$ in the range of $\{2, 4, 6, 8, 10\}$.
(b) **Residual coefficient $\alpha$:** for the residual coefficient in APPNP, we search it in the range of $\{0.1, 0.2\}$.
(c) **Hidden dimension:** We collect the set of dimensions that are widely adopted by existing work as the candidates, i.e., $\{16,32,64,128,256,512\}$.
(d) **Activation function:** The set of available activation functions is listed as follows: {Sigmoid, Tanh, ReLU, Linear, Softplus, LeakyReLU, ReLU6, ELU}

In total, for each dataset we search $480$ architectures of APPNP and we perform the search process on Cora, Citeseer and Ogbn-arxiv. Specifically, we train each architecture on the reduced graph for epochs on as the model converges faster on the smaller graph. We use the best validation accuracy to choose the final architecture. We report (1) the Pearson correlation between validation accuracies obtained by architectures trained on condensed graphs and those trained on original graphs, and (2) the average test accuracy of the searched architecture over 20 runs.

Table 9: Neural Architecture Search. Methods are compared in validation accuracy correlation and test accuracy obtained by searched architecture. Whole means the architecture is searched using whole dataset.

| | Pearson Correlation | | | Test Accuracy | | | |
|---|---|---|---|---|---|---|---|
| | Random | Herding | GCOND | Random | Herding | GCOND | Whole |
| Cora | 0.40 | 0.21 | **0.76** | 82.9 | 82.9 | 83.1 | 82.6 |
| Citeseer | 0.56 | 0.29 | **0.79** | 71.4 | 71.3 | 71.3 | 71.6 |
| Ogbn-arxiv | 0.63 | 0.60 | **0.64** | 71.1 | 71.2 | 71.2 | 71.9 |

## C.3 TIME COMPLEXITY AND RUNNING TIME

**Time Complexity.** We start from analyzing the time complexity of calculating gradient matching loss, i.e., line 8 to line 11 in Algorithm 1. Let the number of MLP layers in $g_\Phi$ be $L$ and $r$ be the number of sampled neighbors per node. For simplicity, we assume all the hidden units are $d$ for all layers and we use $L$-layer GCN for the analysis. The forward process of $g_\Phi$ has a complexity of $O(N'^2d^2)$. The forward process of GCN on the original graph has a complexity of $O(r^L N d^2)$ and that on condensed graph has a complexity of $O(LN'^2 d + LN' d)$. The complexity of calculating the second-order derivatives in backward propagation has an additional factor of $O(|\boldsymbol{\theta}_t||\mathbf{A}'| + |\boldsymbol{\theta}_t||\mathbf{X}'|)$, which can be reduced to $O(|\boldsymbol{\theta}_t| + |\mathbf{A}'| + |\mathbf{X}'|)$ with finite difference approximation. Although there are $C$ iterations in line 7-11, we note that the process is easily parallelizable. Furthermore, the process of updating $\boldsymbol{\theta}_t$ in line 16 has a complexity of $\tau_{\boldsymbol{\theta}}(LN'^2 d + LN' d)$. Considering there are $T$ iterations and $K$ different initializations, we multiply the aforementioned complexity by $KT$. To

sum up, we can see that the time complexity linearly increases with number of nodes in the original graph.

**Running Time.** We now report the running time of the proposed GCOND for different condensation rates. Specifically, we vary the condensation rates in the range of $\{0.1\%, 0.5\%, 1\%\}$ on Ogbn-arxiv and $\{1\%, 5\%, 10\%\}$ on Cora. The running time of 50 epochs on one single A100-SXM4 GPU is reported in Table 10.The whole condensation process (1000 epochs) for generating 0.5% condensed graph of Ogbn-arxiv takes around 2.4 hours, which is an acceptable cost given the huge benefits of the condensed graph.

Table 10: Running time of GCOND for 50 epochs.

| $r$ | 0.1% | 0.5% | 1% | $r$ | 1% | 5% | 10% |
|---|---|---|---|---|---|---|---|
| Ogbn-arxiv | 348.6s | 428.2s | 609.8s | Cora | 37.4s | 43.9s | 64.8s |

### C.4 SPARSIFICATION

In this subsection, we investigate the effect of threshold $\delta$ on the test accuracy and sparsity. In detail, we vary the values of the threshold $\delta$ used for truncating adjacency matrix in a range of $\{0.01, 0.05, 0.1, 0.2, 0.4, 0.6, 0.8\}$, and report the corresponding test accuracy and sparsity in Figure 3. From the figure, we can see that increasing $\delta$ can effectively increase the sparsity of the obtained adjacency matrix without affecting the performance too much.

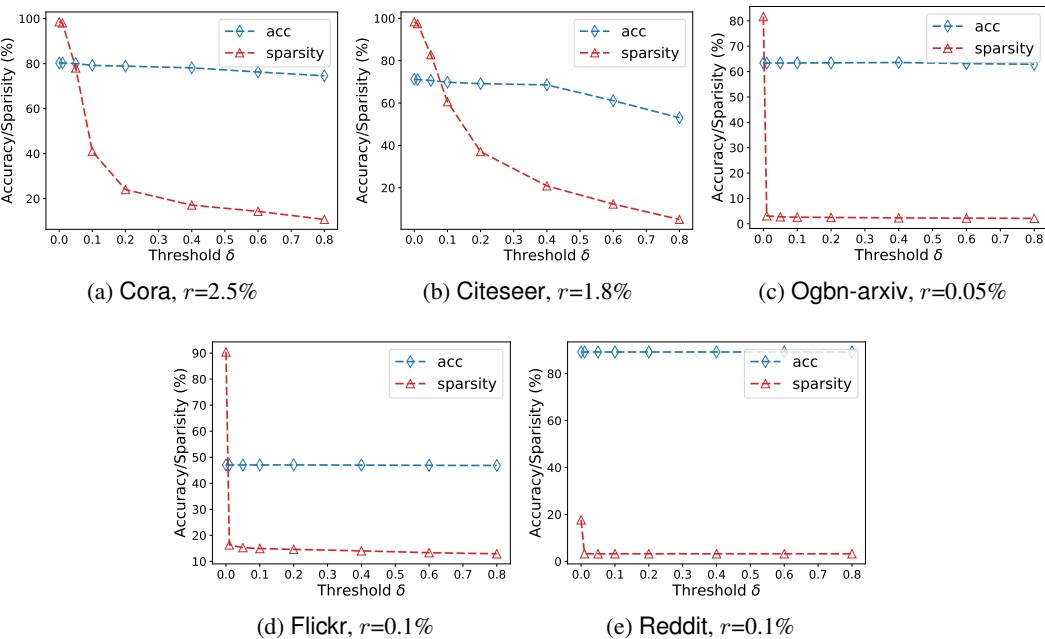

Figure 3: Test accuracy and sparsity under different values of $\delta$.

### C.5 DIFFERENT DEPTH AND HIDDEN UNITS.

**Depth Versus Hidden Units.** We vary the number of model layers (GCN) in a range of $\{1, 2, 3, 4\}$ and the number of hidden units in a range of $\{16, 32, 64, 128, 256\}$, and test them on the condensed graphs of Cora and Citeseer. The results are reported in Table 11. From the table, we can observe that changing the number of layers impacts the model performance a lot while changing the number of units does not.

Table 11: Test accuracy on different numbers of hidden units (H) and layers (L). When L=1, there is no hidden layer so the number of hidden units is meaningless.

<table>
<tr><td colspan="5" align="center">(a) Cora, $r$=2.6%</td><td colspan="5" align="center">(b) Citeseer, $r$=1.8%</td></tr>
<tr><td>H\L</td><td>1</td><td>2</td><td>3</td><td>4</td><td>H\L</td><td>1</td><td>2</td><td>3</td><td>4</td></tr>
<tr><td>16</td><td>74.8±0.5</td><td>76.8±1.0</td><td>68.0±3.0</td><td>50.9±9.5</td><td>16</td><td>58.6±12.1</td><td>69.2±1.3</td><td>56.9±8.4</td><td>40.4±1.2</td></tr>
<tr><td>32</td><td>-</td><td>79.2±0.7</td><td>70.4±3.2</td><td>61.1±7.2</td><td>32</td><td>-</td><td>69.4±1.3</td><td>59.9±10.2</td><td>42.6±3.6</td></tr>
<tr><td>64</td><td>-</td><td>79.2±1.0</td><td>72.0±3.3</td><td>64.5±2.2</td><td>64</td><td>-</td><td>69.7±1.5</td><td>62.3±10.3</td><td>43.6±3.7</td></tr>
<tr><td>128</td><td>-</td><td>79.9±0.3</td><td>76.6±1.8</td><td>61.8±3.8</td><td>128</td><td>-</td><td>70.2±1.4</td><td>63.3±9.7</td><td>51.6±1.8</td></tr>
<tr><td>256</td><td>-</td><td>80.1±0.6</td><td>75.9±1.6</td><td>65.6±2.9</td><td>256</td><td>-</td><td>70.6±0.9</td><td>63.5±10.0</td><td>52.9±5.5</td></tr>
</table>

Table 12: Cross-depth accuracy on Cora, $r$=2.6%

| C\T | 2 | 3 | 4 | 5 | 6 |
|-----|-----|-----|-----|-----|-----|
| 2 | 80.30 | 80.70 | 79.46 | 76.06 | 71.23 |
| 3 | 40.62 | 72.37 | 40.14 | 67.19 | 35.02 |
| 4 | 74.24 | 72.56 | 76.26 | 71.70 | 65.12 |
| 5 | 71.31 | 75.73 | 70.95 | 73.13 | 67.12 |
| 6 | 75.20 | 75.18 | 75.67 | 76.16 | 75.00 |

**Propagation Versus Transformation.** We further study the effect of propagation and transformation on the condensed graph. We use Cora as an example and use SGC as the test model due to its decoupled architecture. Specifically, we vary both the propagation layers and transformation layers of SGC in the range of $\{1, 2, 3, 4, 5\}$, and report the performance in Table 13. As can be seen, the condensed graph still achieves good performance with 3 and 4 layers of propagation. Although the condensed graph is generated under 2-layer SGC, it is able to generalize to 3-layer and 4-layer SGC. When increasing the propagation to 5, the performance degrades a lot which could be the cause of the oversmoothing issue. On the other hand, stacking more transformation layers can first help boost the performance but then hurt. Given the small scale of the graph, SGC suffers the overfitting issue in this case.

**Cross-depth Performance.** We show the cross-depth performance in Table 12. Specifically, we use SGC of different depth in the condensation to generate condensed graphs and then use them to test on SGC of different depth. Note that in this table, we set weight decay to 0 and dropout to 0.5. We can observe that usgin a deeper GNN is not always helpful. Stacking more layers do not necessarily mean we can learn better condensed graphs since more nodes are involved during the optimization, and this makes optimization more difficult.

Table 13: Test accuracy of SGC on different transformations and propagations for Cora, $r$=2.6%

| Trans\Prop | 1 | 2 | 3 | 4 | 5 |
|-----|-----|-----|-----|-----|-----|
| 1 | 77.09±0.43 | 79.02±1.17 | 78.12±2.13 | 74.04±3.60 | 61.19±7.73 |
| 2 | 76.94±0.50 | 79.01±0.57 | 79.11±1.15 | 77.57±1.03 | 72.37±4.25 |
| 3 | 75.28±0.58 | 77.95±0.67 | 74.16±1.50 | 70.58±3.71 | 58.28±8.90 |
| 4 | 66.87±0.73 | 66.54±0.82 | 59.24±1.60 | 43.94±6.33 | 30.45±9.67 |
| 5 | 46.44±0.91 | 37.29±3.23 | 16.05±2.74 | 15.33±2.79 | 15.33±2.79 |

C.6    VISUALIZATION OF NODE FEATURES.

In addition, we provide the t-SNE (Van der Maaten & Hinton, 2008) plots of condensed node features to further understand the condensed graphs. In Cora and Citeseer, the condensed node features are well clustered. For Ogbn-arxiv and Reddit, we also observe some clustered pattern for the nodes within the same class. In contrast, the condensed features are less discriminative in Flickr, which indicates that the condensed structure information can be essential in training GNN.

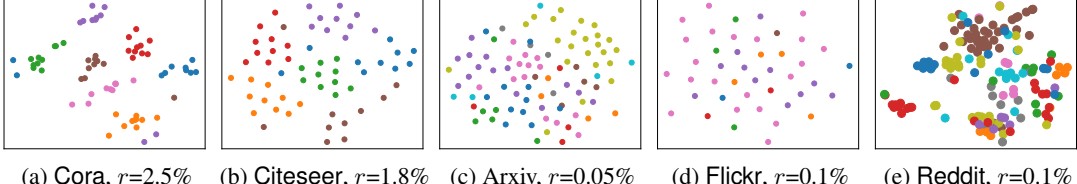

(a) Cora, $r$=2.5%    (b) Citeseer, $r$=1.8%    (c) Arxiv, $r$=0.05%    (d) Flickr, $r$=0.1%    (e) Reddit, $r$=0.1%

Figure 4: The t-SNE plots of node features in condensed graphs.

## C.7 PERFORMANCES ON ORIGINAL GRAPHS

We show the performances of various GNNs on original graphs in Table 14 to serve as references.

Table 14: Performances of various GNNs on original graphs. SAGE: GraphSAGE.

|            | GAT  | Cheby | SAGE | SGC  | APPNP | GCN  |
|------------|------|-------|------|------|-------|------|
| Cora       | 83.1 | 81.4  | 81.2 | 81.4 | 83.1  | 81.2 |
| Citeseer   | 70.8 | 70.2  | 70.1 | 71.3 | 71.8  | 71.7 |
| Ogbn-arxiv | 71.5 | 71.4  | 71.5 | 71.4 | 71.2  | 71.7 |
| Flickr     | 44.3 | 47.0  | 46.1 | 46.2 | 47.3  | 47.1 |
| Reddit     | 91.0 | 93.1  | 93.0 | 93.5 | 94.3  | 93.9 |

## C.8 EXPERIMENTS ON PUBMED.

We also show the experiments for Pubmed with condensation ratio of 0.3% in Table 15. From the table, we can observe that GCOND approximates the original performance very well (77.92% vs. 79.32% on GCN). It also generalizes well to different architectures and outperforms GCOND-X and DC-Graph, indicating that it is important to leverage the graph structure information and learn a condensed structure.

Table 15: Performance of different GNNs on Pubmed ($r$=0.3%).

|          | APPNP       | Cheby       | GCN         | GraphSage   | SGC         |
|----------|-------------|-------------|-------------|-------------|-------------|
| DC-Graph | 72.76±1.39  | 72.66±0.59  | 72.44±2.90  | 71.96±2.50  | 75.43±0.65  |
| GCOND-X  | 73.91±0.41  | 74.57±1.00  | 71.81±0.94  | 73.10±2.08  | 76.72±0.65  |
| GCOND    | 76.77±1.17  | 75.48±0.82  | 77.92±0.42  | 71.12±3.10  | 75.91±1.38  |

## D MORE RELATED WORK

**Graph pooling.** Graph pooling (Zhang et al., 2018; Ying et al., 2018b; Gao & Ji, 2019) also generates a coarsened graph with smaller size. Zhang et al. (2018) is one the first to propose an end-to-end architecture for graph classification by incorporating graph pooling. Later, DiffPool (Ying et al., 2018b) proposes to use GNNs to parameterize the process of node grouping. However, those methods are majorly tailored for the graph classification task and the coarsened graphs are a byproduct graph during the representation learning process.

