# OpenReview forum: "Graph Condensation for Graph Neural Networks"
_ICLR.cc/2022/Conference — ICLR 2022 Poster_

### Official Review · Reviewer_peGb · 2021-10-20

**Correctness:** 3
**Technical Novelty And Significance:** 2
**Empirical Novelty And Significance:** 2
**Recommendation:** 6
**Confidence:** 4

**Main Review:**

Overall, I personally found the idea of the problem presented in the paper intriguing. Being able to shrink large datasets to smaller ones in order to allow an easy training and tuning of a variety of approaches is something that would for sure help both in academic and industrial settings.

This said, I’m not sure the solution proposed in the paper fully addresses the problem. In order to make the new condensed graphs actually useful for training and tuning, we would need a variety of different GNNs to actually approach their original performance when actually trained on the smaller synthetic graphs or at least to show the same order of performance (i.e. if performance of GNN1 are better than GNN2 when trained on the original graph, they should also be better when trained on the synthetic graph), so that we could for instance tune a variety of different architectures on the smaller condensed graphs G’ and then re-train only the best performing one on the original datasets. However, these conditions don’t seem to be satisfied for a variety of approaches. Besides for GCN, which achieves in 4 out of 5 datasets performance which are reasonably close to the ones it would have if trained on the original graphs, the generalizability of condensed graphs to different architectures doesn’t seem particularly strong as outlined in Table 3. For instance, Graph Attention Network which achieves 83% on the original CORA, it achieves only 66.2% on the condensed one with SGC and, in addition to this, it is actually outperformed by GCN which instead achieves lower performance than GAT if trained on the original dataset (a similar, although less marked, comparison could be made for APPNP as well, which is outperformed by GCN both on Citeseer and Cora, while outperforming it if trained on the original datasets as per https://arxiv.org/pdf/1903.02428v3.pdf). This is most likely due to the way in which the condensed graphs are constructed (i.e. via SGC), which intrinsically pushes the condensed graph to perform well for the model used at condensation time (or models which are reasonably similar to this one, e.g. GCN) but it doesn’t offer any guarantee for reasonably different models.

For this reason, while I personally think that the problem outlined in paper is of interest to the community, I’m not sure the solution proposed in the manuscript is mature enough for pushing for an acceptance to ICLR.

In addition to this, there are a variety of other points that I would like to highlight in my review and which I would like the authors to address in their rebuttal:

1) Equation 6, the distance measure boils down to the sum of one minus the cosine similarity between the columns of the two different gradients. Why did the authors choose that specific distance and not for instance an L2 distance? What is the effect of that distance on performance of the approach?

2) In the graph sampling paragraph of subsection 3.1, the authors state “To further reduce memory usage and ease optimization, we calculate the gradient matching loss for nodes from different classes separately, as matching the gradients w.r.t. the data from a single class is easier than that from all classes.”, however there is not really an explanation in the paper why this would be easier. Do the authors match gradients on each class separately as they want nodes of the same class to provide the same information? why? I believe the approach should work even if we aggregate the gradients on all samples from different classes.

3) In the same subsection as for points 2, the authors also state “For the condensed graph A’, we sample a batch of synthetic nodes of class c but do not sample their neighbors as we need to learn the connections with other nodes.”. Does this imply that their model will learn connections only across the sampled nodes at each step instead of out of the entire pool of available nodes?

4) The authors evaluate their approach on Cora and Citeseer (among the other datasets) but they didn’t consider PubMed which is typically used in experimental evaluations alongside these two. Is there a specific reason why the authors decided to skip PubMed in their experimental evaluation?

5) In Section 4.1 the authors state “As vanilla DC cannot leverage any structure information, we develop a variant named DC-Graph, which additionally leverages graph structure during test stage, to replace DC for the following experiments.”. However, as for DC-Graph the authors use only the identity matrix on the condensed graph at training time, while they use the whole original graph at test time, it is unclear how DC and DC-Graph differ at test time. Did the authors discard the connectivity of the original graph during testing when evaluating a model trained on graph condensed with DC? If that is correct, why?

6) Section 4.3, paragraph “Different Architectures” is quite unclear. I believe the authors are showing the performance that different architectures tested with graphs condensed with SGC allow to achieve, however these numbers don't match the ones reported in table 4 for ChebNet. I believe this is just a typo and I would ask the authors to please confirm this and if I'm correct to fix the error in the table. Additionally, I would kindly ask the authors to please report the performance that the considered models were able to achieve on the original graphs. It would greatly help at highlighting the variation in performance of the approaches wrt the ideal scenario (i.e. where we train and test them on the same graph)

7) Section 4.3, paragraph “Versatility of GCOND”, the authors omit GAT out of the comparison showing the performance of a graph condensed with a GNN but then used for training with a different one because of its “deterioration in performance with large neighborhood sizes”. However, this doesn’t appear to me as a sufficient justification for omitting GAT altogether and if this fails at performing well in those specific conditions it should probably be shown in the table with the other methods.  Additionally, couldn't we restrict the neighborhood sizes in the condensed graph (e.g. by considering only the K most similar neighbors) before training GAT if this fails to perform well with large neighborhoods?

8) In Table 4, surprisingly a graph condensed with GCN is not the best one for training GCN (the graph condensed with SGC allows to achieve 10% more in terms of accuracy with GCN). This is quite surprising and I wonder if the authors have a justification for this.

9) Table 5, the graph condensed with SGC for OGBN-Arxiv loses its original homophily. This is probably the reason for the drop in performance, however I was not able to identify in the paper a reason for this. Do the authors have a justification for this behavior?

----

Updated score to 6 after author's rebuttal.

**Summary Of The Paper:**

The paper proposes and addresses the problem of graph condensation. In a nutshell, provided a large graph G, the scope of the paper is to propose a solution able to generate a smaller synthetic graph G’, which effectively allows to train Graph Neural Networks (GNNs) able to achieve similar performance as if they were trained on G. In order to do that, the authors propose to match the gradients produced by a specific GNN on mini-batches extracted from G and G’. This avoids the need of constructing a minimization problem based on an outer and inner training loop which would be computationally heavy to handle. The node-wise features X’ of the smaller graph G’ are considered as free parameters in the optimization process and the adjacency matrix A’ is instead inferred from X’ via a learnable MLP which is trained alongside X’ with an alternating optimization schema. The authors evaluate the quality of their approach on the Cora, Citeseer, OGBN-Arxiv and Reddit datasets, condensing and evaluating the new graphs with a variety of different architectures.

**Summary Of The Review:**

The paper proposes and addresses the problem of graph condensation. While the formulation of the problem is interesting, I didn’t find the approach proposed by the authors to be mature enough for ICLR due to the limitations and issues highlighted in the main review. I thus consider the paper a weak reject for the conference and I kindly ask the authors to reply to the points highlighted in my review in their rebuttal.

---

> ### Author Response · Authors · 2021-11-23
> **Response to Reviewer peGb (1/3)**
>
> Thank you for your very detailed comments and useful suggestions. It greatly helps us to improve this work.
>
> > Q1. The condensed data does not seem mature for tuning a variety of different architectures on the smaller condensed graphs and then re-train only the best performing one on the original datasets.
>
> A1. We appreciate your attention to detail on this point, and included the updated experiments which show that the condensed graphs are shown to be useful for training and tuning in the revised draft. Specifically, we perform experiments on neural architecture search for a single reference model, APPNP, in Table 9 in Appendix B.2. We observe a very high correlation between the validation accuracies obtained by architectures trained on condensed graphs and those trained on original graphs. For example, on Citeseer, the Peason correlation between the validation accuracies is 0.76 while Herding and Random only achieve 0.40 and 0.21, respectively. Such results show that we are actually able to tune a variety of different architectures on the smaller condensed graph.
>
> > Q2. The generalizability of condensed graphs to different architectures doesn’t seem particularly strong as outlined in Table 3. For instance, GAT and APPNP as well, which is outperformed by GCN both on Citeseer and Cora. This is most likely due to the way in which the condensed graphs are constructed (i.e. via SGC), which intrinsically pushes the condensed graph to perform well for the model used at condensation time but it doesn’t offer any guarantee for reasonably different models.
>
> A2. Thank you for this great question. We would like to draw your attention to the following points:
> *  **Slight hyper-parameter tuning can boost the performance.** We would like to point out that the performance of APPNP in Table 3 is obtained from a 2-layer APPNP. We are actually able to further improve the performance by tuning its architecture hyper-parameters (e.g., number of propagations $K$). Now we set the number of APPNP to be 10 (the default setting in the APPNP paper).  We can obtain an accuracy of 81.7% (corresponding to 83.1% obtained from the original graph), which is better than the GCN performance (80.1%) on the same condensed graph.  As for GAT, we believe it is more complicated and different and we will further discuss it in the following bullets.
> * **Inductive biases among architectures.** Intuitively, matching two sets of gradients w.r.t. two architectures enables the condensed graph to encode discriminative information for the target task. The condensed graphs capture the correlation between original graph and high-dimensional deep representations in a very compressed way. Such correlations are informative to the architectures sharing similar inductive biases. The DC paper [1] also suggests a similar notion. In our case, during the forward process, GCN, SGC, APPNP, GraphSAGE and Cheby all use predefined propagation matrices (e.g., augmented normalized adjacency) to aggregate information from neighbors, while GAT learns propagation coefficients (or attention score) in a data-driven manner.  Hence, the inductive bias between SGC and GAT is less similar to that between SGC and GCN.  As a result, our experiments show that condensed graphs learned in other GNNs do not generalize well to GAT but they work well for GCN, SGC, APPNP, GraphSAGE which share more similar inductive biases.
> * **Further notes.** Empirically, we expect the condensed data to be more applicable/useful for models with similar inductive bias (similar line of argument to dataset condensation paper).  We anticipate that as a result, condensation from same model to same (or similar) model is expected to do better.  General-purpose (unbiased by the condenser) is a very appealing direction with more implications for cross-architecture comparisons, but it is quite challenging and non-trivial, especially under the gradient matching framework, hence we leave it for future work.
>
>
> > Q3.  Why did the authors choose that specific distance (Eq. 6) and not for instance an L2 distance? What is the effect of that distance on performance of the approach?
>
> A3. Equation (6) has been shown to be empirically more effective than other distance functions in [1]. We followed this convention and found it worked well in the graph case. We can also replace the gradient matching loss with L2 distance. Below we show the experimental results of using L2 distance:
>
> || Cora, r=2.6%| Citeseer, r=1.8%|Ogbn-arxiv, r=0.05%|Flickr, 0.5%|Reddit, 0.1%|
> |--|--|--|--|--|--|
> |L2 distance|79.6±0.4|68.3±1.2|57.0±1.5|47.0±0.1|88.7±0.5|
> |Ours|80.1±0.6|70.6±0.9|59.2±1.1|47.1±0.1|89.6±0.7|
>
> The loss used in GCond achieves better performances in all settings, suggesting that it is more effective and robust than L2 distance in the graph condensation problem.

---

> > ### Author Response · Authors · 2021-11-23
> > **Response to Reviewer peGb (2/3)**
> >
> > > Q4. In graph sampling , why do the authors match gradients on each class separately?
> >
> > A4.  We match gradients on each class and update each class separately. The intuition is that imitating the mean gradients w.r.t. the data from a single class is easier compared to those of multiple classes as the information is less mixed.  This is also the convention adopted in the DC paper [1].  Our experiments in the the following table also demonstrate that it yields better test accuracy.
> >
> > |    **Table of Test Accuracy**                           | Cora, r=2.6%   | Citeseer, r=1.8%   |
> > |-----|---|----|
> > | Match different classes       | 62.4 ± 5.6 | 32.1 ± 8.2 |
> > | Ours (match one single class) | 80.1 ± 0.6 | 70.6 ± 0.9 |
> >
> > Moreover, the table below shows the loss at the first epoch and final epoch for both methods. For example, 0.96/0.54 indicates that the initial loss is 0.97 while the  loss at the final epoch  is 0.54. We find that matching only one single class is indeed easier as it results in much greater loss reduciton.
> >
> > |      **Table of Loss Reduciton**       | Cora, r=2.6% | Citeseer, r=1.8% |
> > |-----|---|---|
> > | Match different classes       | 0.97/0.54    | 0.95/0.69        |
> > | Ours (match one single class) | 2.2/0.15     | 1.9/0.19         |
> >
> > Moreover,  we find that  the gradient matching loss change on Cora: loss condensed with SGC can drop 99% while loss condensed with GCN can only drop less than 50%.
> >
> > > Q5. The authors also state “For the condensed graph A’, we sample a batch of synthetic nodes of class c but do not sample their neighbors as we need to learn the connections with other nodes.” Does this imply that their model will learn connections only across the sampled nodes at each step instead of out of the entire pool of available nodes?
> >
> > A5. No, we learn connections between sampled nodes and all other available nodes in the entire pool. The statement “do not sample their neighbors” means that we will aggregate information from all other nodes at the forward step. To make it clear, we have updated the description in Section 3.1.
> >
> > > Q6. Results on Pubmed.
> >
> > A6.  The reason why we did not try Pubmed in the first place is due to its extremely low labeling rate (around 0.3%). Per your suggestion, we now apply the proposed GCond on Pubmed dataset and report the performance in the following table (we have also included it in Table 15 in Appendix B.7):
> >
> >   |          | APPNP      | Cheby      | GCN        | GraphSage  | SGC        |
> > |---|---|--|---|---|---|
> > | DC-Graph | 72.76±1.39 | 72.66±0.59 | 72.44±2.90 | 71.96±2.50 | 75.43±0.65 |
> > | GCond-X  | 73.91±0.41 | 74.57±1.00 | 71.81±0.94 | 73.10±2.08 | 76.72±0.65 |
> > | GCond    | 76.77±1.17 | 75.48±0.82 | 77.92±0.42 | 71.12±3.10 | 75.91±1.38 |
> >
> > From the table, we can observe that GCond also approximates the original performance of Pubmed very well (77.92% vs. 79.32% on GCN). It also generalizes well to different architectures.
> >
> > > Q7. As for DC-Graph the authors use only the identity matrix on the condensed graph at training time, while they use the whole original graph at test time, it is unclear how DC and DC-Graph differ at test time. Did the authors discard the connectivity of the original graph during testing when evaluating a model trained on graph condensed with DC? If that is correct, why?
> >
> > A7. The only difference between DC and DC-Graph happens at test time: DC only uses node features to do inference while DC-Graph uses both features and connectivity of the original graph. The vanilla DC is designed for image data and it does not take advantage of the graph structure at test time. So, here we made a simple adjustment to adapt DC to the graph domain and denote it as DC-Graph. We kindly refer the reviewer to Table 1, where we show the information. We also copy part of the table as follows.
> >
> > |              | DC        | DC-Graph        |
> > |---|---|--|
> > | Condensation | $X_\text{train}$ |  $X_\text{train}$  |
> > | Training     | $X'$     | $X'$              |
> > | Test         |  $X_\text{test}$     | $A_\text{test}$,    $X_\text{test}$ |
> >
> > > Q8. These numbers in Table 3 don't match the ones reported in table 4 for ChebNet. I believe this is just a typo and I would ask the authors to please confirm this and if I'm correct to fix the error in the table.
> >
> > A8. Thank you for pointing it out. It is a typo and we have corrected it.
> >
> > > Q9. Additionally, I would kindly ask the authors to please report the performance that the considered models were able to achieve on the original graphs. It would greatly help at highlighting the variation in performance of the approaches w.r.t. the ideal scenario (i.e. where we train and test them on the same graph).
> >
> > A9. Thank you for your suggestions. Due to the page limit of main content of the paper,  we have included them in Table 14 of the Appendix B.6.

---

> > > ### Author Response · Authors · 2021-11-23
> > > **Response to Reviewer peGb (3/3)**
> > >
> > >
> > > > Q10. Missing performance of GAT in Table 4
> > >
> > > A10. We provide our response from the following two aspects:
> > > * During the evaluation stage, we agree that we can report the performance of GAT as the test model. Per your suggestion, we add the experiments on including GAT  as the test model in the following table. We can observe that the condensed graphs from other GNNs also show certain generalization performance on GAT.
> > >
> > > | Dataset    | T\C | APPNP1     | Cheby      | GCN        | GraphSage  | SGC|
> > > |------------|-----|------------|------------|------------|------------|---|
> > > | Ogbn-arxiv | GAT | 55.41±3.06 | 56.29±0.94 | 54.38±1.43 | 53.75±2.20 | 60.05±1.82 |
> > > | Cora       | GAT | 67.53±1.30 | 70.12±4.22 | 64.31±3.15 | 56.00±9.15 | 66.20±3.28|
> > >
> > > * However, during the condensation process, we need to learn a dense graph adjacency matrix, which is hard for GAT to handle given the large number of node pairs ($N'^2$ attention scores for one single attention head) [2]. If we sample edges/entries from the dense matrix and feed them into GAT, the unsampled entries would not be updated and thus affect the convergence of the framework. Hence, we decide not to include GAT as the model used in condensation.
> > >
> > >
> > > > Q11. In Table 4, surprisingly a graph condensed with GCN is not the best one for training GCN (the graph condensed with SGC allows to achieve 10% more in terms of accuracy with GCN). This is quite surprising and I wonder if the authors have a justification for this.
> > >
> > > A11. Indeed, we observe that the graph condensed with GCN may not necessarily be the best one for training GCN in practice. In practice, we observe that for simpler datasets (e.g., Cora and Citeseer), simpler models (e.g., SGC with one transformation layer) used in condensation greatly ease the optimization of the condensed graph. This is because simpler models involve less network parameters and thus smaller size of gradients, which are less difficult to imitate. As a result, the graph condensed from SGC shows competitive generalization performance across different architectures compared to  the one condensed with GCN (with two transformation layers). To see this more clearly, we show the gradient matching loss change on Cora: loss condensed with  SGC can drop 99% while loss condensed with GCN can only drop less than 50%.
> > >
> > > > Q12. Table 5, the graph condensed with SGC for OGBN-Arxiv loses its original homophily. This is probably the reason for the drop in performance, however I was not able to identify in the paper a reason for this. Do the authors have a justification for this behavior?
> > >
> > > A12.
> > > * While it is very difficult to provide a definite answer for the question, our intuition is that the optimization process for learning condensed graphs is not constrained with any regularization terms. Hence, it does not provide guarantees on the preservation of homophily.  Since the features are also learned along with the condensed graph while using the original training data’s gradient information, it is also possible for some of the homophily in the original context to be made redundant/unnecessary in the condensed graph, if the features are learned appropriately. Intuitively, consider a case where the mutual information between a node’s own label and its 1-hop neighbor features is extremely high; although the original graph may be strongly homophilous, it’s possible for these 1-hop neighbor features to be “absorbed” into the condensed sample features and make the need for learned homophily in the condensed graph unimportant.
> > >
> > >   However, it is indeed possible that “losing the homophily” is a cause for failing  to reproduce the original accuracy. To address such issues, we can add various priors,  e.g. homophily regularization to force the condensed graph to encode some homophily information.  In this specific setting, with the homophily regularization, we further increase the condensation ratio from 0.5% to 1% and we finally obtain 66.1% accuracy, which is 66.1/71.4=92.6% of the original performance.
> > >
> > > * It is worth noting that even in our presented case of “lost homophily”, the performance of GCond is already quite competitive and outperforms the other baselines such as Random, Herding, etc. by a large margin despite this detail.
> > >
> > >
> > > Thank you again for taking the time to review our paper. We hope our responses could clarify your concerns, and hope you will consider increasing your score. If we have left any notable points of concern unaddressed, please do share and we will attend to these points.
> > >
> > > ---
> > > [1] Zhao et al. Dataset condensation with gradient matching. ICLR 2021. \
> > > [2] Ma et al. Learning discrete adaptive receptive fields for graph convolutional networks. OpenReview 2021.

---

### Official Review · Reviewer_XqrK · 2021-11-02

**Correctness:** 4
**Technical Novelty And Significance:** 2
**Empirical Novelty And Significance:** 4
**Recommendation:** 5
**Confidence:** 4

**Details Of Ethics Concerns:**

There is no foreseeable ethics concern with this paper.

**Main Review:**

Strength:
1. This paper is generally well-written and easy to follow.

2. The paper's motivation is clear.

3. Extensive experiments are conducted to demonstrate the model's effectiveness.

Weakness:
1. This paper attempts to extend the Dataset Condensation model and Gradient Matching technique presented in [1] to the domain of GNNs. Although some differences exist, they are so minor that the main equations (1) - (5) in this work are nearly identical to equations (4), (5), and (7) - (9) in [1]. As a result, I believe this work lacks innovation in terms of technical contribution.

[1] Bo Zhao, Konda Reddy Mopuri, and Hakan Bilen. Dataset condensation with gradient matching. https://openreview.net/pdf?id=mSAKhLYLSsl.


2. In Section 3.2, the authors claim that GCond-X, which only learns the condensed node features X' and not the condensed graph structure A', also performs well. This observation leads me to wonder if the proposed method is better suited for *dataset* condensation rather than *graph* condensation.

3. The paper makes no attempt to analyze the algorithm's complexity or to report its runtime.

Minor comments:
1. In equation (3), the letter 'D' should be in the mathematical font.

2. On page 4, line 1, it appears that the phrases '[...] and unrolling the entire training trajectory of the inner problem' have been omitted or repeated.

3. Page 6, Evaluation section, line 3: 'With r percent of N nodes'. Here r should be a number between 0 and 100, and yet it is a percentage in Table 2.

4. All 'r=...' in the supporting materials appear to be in a non-math typeface.

**Summary Of The Paper:**

This paper introduces GCond, a graph condensation framework designed to compress graph datasets and reduce storage and time requirements while training GNNs on large-scale graphs. GCond makes use of gradient matching and graph sampling to reduce the graph size. Experiments demonstrate that GCond can maintain a high degree of accuracy while significantly lowering the size of the graph.

**Summary Of The Review:**

The paper's idea is interesting, as evidenced by the experimental results. However, the theoretical depth and novelty may not be enough to meet ICLR's standards.

---

> ### Author Response · Authors · 2021-11-23
> **Response to Reviewer XqrK  (1/2)**
>
> Thank you for the detailed comments and valuable questions! Your comment of “the paper's idea is interesting” is very encouraging! Next we provide details to clarify your major concerns.
>
> > Q1. Technical contributions of this work.
>
> A1. We thank the reviewer, and will clarify our technical novelty and significance during revision.
>
> (1) First, as the first attempt on condensing graph datasets, it is **nontrivial** to develop the dataset condensation framework that shifts from images to graphs, since most components need to be adapted:
> * Modeling the dependencies between nodes.
>   * Instead of directly learning A’ and X’ as free parameters,  we show that *parameterization of graph structure and node features* can better handle the dependencies between nodes and achieve up to 8% improvement as shown in our ablation study in Table 7.   Moreover, *such parameterization better scales*, because the number of parameters does not grow quadratically with the number of condensed nodes.
>
>   * The *optimization scheme* also matters, as shown in Table 8. We turn to training the graph structure and node features alternatively. It resembles the real scenarios where node features affect the connectivities and then the connectivities also affect the node features in turn.
> * We also design the *sampling strategies* to enable efficient condensation and a *sparsification strategy* to further reduce the storage of condensed graphs.
> * Our empirical performances are very significant. We are able to approximate the original test accuracy by 99% on three datasets, 95% on one dataset and 92% on the remaining dataset, with more than 99% reduction of graph size.
>
> All the aforementioned points are **unique in the graph domain** and not necessarily straightforward to adapt from the image domain.
>
> (2) Second, condensed datasets can be **a useful point of investigation for the graph ML community**. We will open source the learned condensed datasets for further investigation. Particularly, our work yields several interesting followup directions and questions:
> * It may reveal insights into interpretability and the nature of sample-efficient representations. For example, as shown in Table 5 and Figure 2, the condensed graph provides a plausible interpretation of the original dataset and in some cases it preserves the homophily property of the original dataset, though not always.
> * Understanding the importance of preserving these properties (or good performance albeit not preserving them) is an interesting area of further research.
> * The condensed graphs can be used in many critical applications especially for neural architecture search (NAS) and continual learning. In the **NAS experiments** in Appendix B.2, we observe reliable correlation of performances between condensed dataset training and whole-dataset training.
>
> We believe this is an interesting and insightful work and hope you could reevaluate the contributions of our work.
>
> > Q2. In Section 3.2, the authors claim that GCond-X, which only learns the condensed node features X' and not the condensed graph structure A', also performs well. This observation leads me to wonder if the proposed method is better suited for dataset condensation rather than graph condensation.
>
> A2. (1) GCond-X also uses graph structure during condensation. We kindly refer the reviewer to Table 1, which shows that GCond-X takes both graph structure and node features as inputs during the condensation process, while only outputting the condensed node features.
>
> (2) While GCond-X performs well in some cases, learning the condensed graph structure is still very important due to the following reasons:
> * **Visualization and interpretation.** As shown in Table 5 and Figure 2, the condensed graph provides a plausible interpretation of the original dataset and in some cases it preserves the homophily property of the original dataset.
> * **Training some specific GNNs.** Some special GNNs such as GAT require learning additional parameters (e.g., attention mechanism) based on the original connectivity. Without graph structure, we cannot train those special GNNs.
> * **Applications such as NAS.** The condensed graph structure is also helpful in neural architecture search where we need to search numerous GNN architectures. Without the graph structure, one is not able to search the number of propagations or layers in GNNs.

---

> > ### Author Response · Authors · 2021-11-23
> > **Response to Reviewer XqrK (2/2)**
> >
> > >  Q3. Attempts to analyze the algorithm's complexity or to report its runtime.
> >
> > A3. Per your suggestion, we now add time complexity analysis and running time of the condensation process in Appendix B.3. Specifically, we provide the following responses:
> >
> > * **Time complexity.**  We start from analyzing the time complexity of calculating gradient matching loss, i.e., line 8 to line 11 in Algorithm 1. Let the number of MLP layers in $g_{\Phi}$ be $L$ and $r$ be the number of sampled neighbors per node. For simplicity, we assume all the hidden units are $d$ for all layers and we use $L$-layer GCN for the analysis. The forward process of $g_{\Phi}$ has a complexity of $O({N'}^2d^2)$. The forward process of GCN on the original graph has a complexity of $O(r^LNd^2)$ and that on condensed graph has a complexity of $O(LN'^2d+LN'd)$. The complexity of calculating the second-order derivatives in backward propagation has an additional factor of $O(|\boldsymbol{\theta}_t|^2)$, which can be reduced to  $O(|\boldsymbol{\theta}_t|)$  with approximated Hessian-vector products.
> >
> >   Although there are $C$ iterations in line 7-11, we note that the process is easily parallelizable. Furthermore, the process of updating $\boldsymbol{\theta_t}$ in line 16 has a complexity of $\tau_{\boldsymbol{\theta}}(LN'^2d+LN'd)$. Considering there are $T$ iterations and $K$ different initializations, we multiply the aforementioned complexity by $KT$. To sum up, we can see that the time complexity linearly increases with number of nodes in the original graph.
> > * **Table of running time.** We report the running time of the proposed GCond for different condensation rates. Specifically, we vary the condensation rates in the range of {0.1%, 0.5%, 1%} on Ogbn-arxiv and {1%, 5%, 10%} on Cora. The running time of 50 epochs on one single A100-SXM4 GPU is reported in Table 10. Below we also show the running time on the Ogbn-arxiv dataset. The whole condensation process (1000 epochs) for generating 0.5% condensed graph of Ogbn-arxiv takes around 2.4 hours, which is an acceptable cost given the huge benefits of the condensed graph.
> >
> >
> > | Ratio ($r$)      | 0.1%   | 0.5%   | 1%     |
> > |------------|--------|--------|--------|
> > | Ogbn-arxiv | 348.6s | 428.2s | 609.8s |
> >
> > * We note that **we only need to condense the dataset once**. Once condensed, the condensed graphs can be used for training different GNNs and various applications such as neural architecture search and continual learning. Compared to the time and resource expenditure that can be potentially saved by operating on the condensed graphs repeatedly (and only condensing once), the time used in condensation amortizes quickly.
> >
> > ----
> > Thank you again for your constructive comments. We hope our answers would clarify your concerns and please let us know if you have any further questions.

---

### Official Review · Reviewer_R5cV · 2021-11-02

**Correctness:** 3
**Technical Novelty And Significance:** 3
**Empirical Novelty And Significance:** 3
**Recommendation:** 6
**Confidence:** 3

**Main Review:**

Strengths:
1. The idea of using data condensation/distillation on GNNs is new and interesting.
2. The empirical results looks quite promising. It is impressive that the compression ratio can be more than 100, and the testing accuracies remain competitive in many cases.
3. Empirical results show that the synthetic graph constructed based on SGC work well across different models. This generalizability is a useful property of this technique.
4. The paper is well-written.

Weaknesses:
1. Although applying condensation techniques on GNNs is new and some technical details need to be taken care of, the main ideas still follow previous work, e.g. the gradient matching loss.
2. The aim of graph condensation is to alleviate the storage and time consumption. However, to construct the smaller graph, one still needs to train a GNN model on the original graph, which still requires sampling methods for large graphs. I think the authors should provide more discussions on the computation costs of the condensation process.
3. On more complicated datasets such as OGB-arxiv, the accuracy gap is still quite noticeable. And it seems difficult to improve the accuracy by using larger condensed graph size, since it would become more difficult to train the parameters in the synthetic graph.

**Summary Of The Paper:**

To alleviate the storage and time consumption for training GNN models on large graphs, the paper studies graph condensation, which draws inspirations from data distillation/condensation. It first constructs a much smaller synthetic graph, and then train GNN models on this small graph. Empirical results show that, graph condensation can reduce the size of the graph by 99% but still achieves comparable accuracy.

**Summary Of The Review:**

Overall, I think graph condensation is a new and interesting direction, and the empirical results presented in the submission are generally encouraging. On the other hand, given previous work on data condensation, the novelty of the current method is not significant.

---

> ### Author Response · Authors · 2021-11-23
> **Response to Reviewer R5cV (1/2)**
>
> Thank you for the detailed comments and valuable questions! Your comment of “the idea is interesting and the empirical results look promising” is very encouraging! Next we provide details to clarify your major concerns.
>
> > Q1. Novelty of this work.
>
> A1. We thank the reviewer, and will clarify our technical novelty and significance during revision.
>
> (1) First, as the first attempt on condensing graph datasets, it is **nontrivial** to develop the dataset condensation framework that shifts from images to graphs, since most components need to be adapted:
> * Modeling the dependencies between nodes.
>   * Instead of directly learning A’ and X’ as free parameters,  we show that *parameterization of graph structure and node features* can better handle the dependencies between nodes and achieve up to 8% improvement as shown in our ablation study in Table 7.   Moreover, *such parameterization better scales*, because the number of parameters does not grow quadratically with the number of condensed nodes.
>
>   * The *optimization scheme* also matters, as shown in Table 8. We turn to training the graph structure and node features alternatively. It resembles the real scenarios where node features affect the connectivities and then the connectivities also affect the node features in turn.
> * We also design the *sampling strategies* to enable efficient condensation and a *sparsification strategy* to further reduce the storage of condensed graphs.
> * Our empirical performances are very significant. We are able to approximate the original test accuracy by 99% on three datasets, 95% on one dataset and 92% on the remaining dataset, with more than 99% reduction of graph size.
>
> All the aforementioned points are **unique in the graph domain** and not necessarily straightforward to adapt from the image domain.
>
> (2) Second, condensed datasets can be **a useful point of investigation for the graph ML community**. We will open source the learned condensed datasets for further investigation. Particularly, our work yields several interesting followup directions and questions:
> * It may reveal insights into interpretability and the nature of sample-efficient representations. For example, as shown in Table 5 and Figure 2, the condensed graph provides a plausible interpretation of the original dataset and in some cases it preserves the homophily property of the original dataset, though not always.
> * Understanding the importance of preserving these properties (or good performance albeit not preserving them) is an interesting area of further research.
> * The condensed graphs can be used in many critical applications especially for neural architecture search (NAS) and continual learning. In the **NAS experiments** in Appendix B.2, we observe reliable correlation of performances between condensed dataset training and whole-dataset training.
>
> We believe this is an interesting and insightful work and hope you could reevaluate the contributions of our work.

---

> > ### Author Response · Authors · 2021-11-23
> > **Response to Reviewer R5cV (2/2)**
> >
> > > Q2. More discussions on the computation costs of the condensation process.
> >
> > A2. Per your suggestion, we now add time complexity analysis and running time of the condensation process in Appendix B.3. Specifically, we provide the following responses:
> >
> > * **Time complexity.**  We start from analyzing the time complexity of calculating gradient matching loss, i.e., line 8 to line 11 in Algorithm 1. Let the number of MLP layers in $g_{\Phi}$ be $L$ and $r$ be the number of sampled neighbors per node. For simplicity, we assume all the hidden units are $d$ for all layers and we use $L$-layer GCN for the analysis. The forward process of $g_{\Phi}$ has a complexity of $O({N'}^2d^2)$. The forward process of GCN on the original graph has a complexity of $O(r^LNd^2)$ and that on condensed graph has a complexity of $O(LN'^2d+LN'd)$. The complexity of calculating the second-order derivatives in backward propagation has an additional factor of $O(|\boldsymbol{\theta}_t|^2)$, which can be reduced to  $O(|\boldsymbol{\theta}_t|)$  with approximated Hessian-vector products.
> >
> >  Although there are $C$ iterations in line 7-11, we note that the process is easily parallelizable. Furthermore, the process of updating $\boldsymbol{\theta_t}$ in line 16 has a complexity of $\tau_{\boldsymbol{\theta}}(LN'^2d+LN'd)$. Considering there are $T$ iterations and $K$ different initializations, we multiply the aforementioned complexity by $KT$. To sum up, we can see that the time complexity linearly increases with number of nodes in the original graph.
> > * **Table of running time.** We report the running time of the proposed GCond for different condensation rates. Specifically, we vary the condensation rates in the range of {0.1%, 0.5%, 1%} on Ogbn-arxiv and {1%, 5%, 10%} on Cora. The running time of 50 epochs on one single A100-SXM4 GPU is reported in Table 10. Below we also show the running time on the Ogbn-arxiv dataset. The whole condensation process (1000 epochs) for generating 0.5% condensed graph of Ogbn-arxiv takes around 2.4 hours, which is an acceptable cost given the huge benefits of the condensed graph.
> >
> > | Ratio ($r$)      | 0.1%   | 0.5%   | 1%     |
> > |------------|--------|--------|--------|
> > | Ogbn-arxiv | 348.6s | 428.2s | 609.8s |
> >
> > * We note that **we only need to condense the dataset once**. Once condensed, the condensed graphs can be used for training different GNNs and various applications such as neural architecture search and continual learning. Compared to the time and resource expenditure that can be potentially saved by operating on the condensed graphs repeatedly (and only condensing once), the time used in condensation amortizes quickly.
> >
> > > Q3. On OGB-arxiv, the accuracy gap is still quite noticeable. And it seems difficult to improve the accuracy by using larger condensed graph size, since it would become more difficult to train the parameters in the synthetic graph.
> >
> > A3. (1) Firstly, it is worth noting that even there exist some accuracy gap, the performance of GCond is already quite competitive and outperforms the other baselines such as Random, Herding, etc. by a large margin despite this detail.
> >
> > (2) We agree that "it would become more difficult to train the parameters in the synthetic graph" when increasing the condensed graph size. The current optimization process of condensed graphs is not constrained with any regularization terms. We can exert some priors, e.g. homophily regularization to force the condensed graph to encode some homophily information and constrain the search space of solutions.
> > On a separate note, as shown in Table 5, the condensed graph of Ogbn-arxiv loses its original homophily, which could be the reason that the condensed graph loses its original homophily. The added homophily regularization can also help address this issue. In this specific setting, together with the homophily regularization, we further increase the condensation ratio from 0.5% to 1% and we finally obtain 66.1% accuracy, which is 66.1/71.4=92.6% of the original performance.
> >
> >
> > ----
> > Thank you again for your constructive comments. We hope our answers would clarify your concerns and please let us know if you have any further questions.

---

### Official Review · Reviewer_cTj2 · 2021-11-03

**Correctness:** 3
**Technical Novelty And Significance:** 2
**Empirical Novelty And Significance:** 3
**Recommendation:** 6
**Confidence:** 4

**Main Review:**

Strength:
- The paper is easy to follow and overall well-organized.
- The method is sound.
- Various experiments are conducted, including the generalization experiments

Weaknesses:
- The overall method is similar to the existing gradient matching method. The innovation is limited, but the setting is new for applying this method.
- Compared to the existing gradient matching method for data condensation, the major difference in graph problem is the new parameterization for the adjacency matrix A' and X'. The authors claim that it is important to parameterize A' as a function of X'. However, experiments are not conducted to support this very important claim.

Questions:
- What if A' and X' are treated as free parameters? What if A' is independent of X'? For example, we can set $A_{ij}'= relu( W_{ij} - \delta )$ where $W_{ij}$ is a free parameter to replace equation (7). What's the performance of this approach, in terms of both the optimization efficiency and the test performance?
- Experiments use GNNs with 2 layers. What if more layers are used? Intuitively, using fewer layers (e.g., 0 layer in the extreme case) will reduce this problem to the normal data condensation problems on non-graph data. What if a deeper GNN is used so that the graph structure plays a more important role?
- Some statements are not clear enough. For example, for the $\theta_t$ in equation (5), it only mentions this replaces the $\theta_t^S$ and $\theta_t^T$ (I believe there is a typo under equation (5), too). However, how $\theta_t$ is computed is not mentioned. Taking the average of $\theta_t^S$ and $\theta_t^T$?


**Summary Of The Paper:**

This paper studies the graph condensation problem for GNN training, in which a large graph is condensed into a small synthetic graph, and the GNN trained on the small graph is expected to perform similar to the GNN trained on the large graph.

Technically, the overall framework follows the gradient matching method, but specific parameterization is proposed for optimizing the small synthetic graph.


**Summary Of The Review:**

- Pro: The method is sound.
- Con: The method is similar to the existing gradient matching method, but the graph setting is new.
- Con: Experiments can be improved:
-- Ablation study is needed to verify the effectiveness of the proposed parameterization.
-- Experiments on deeper GNNs can better demonstrate the method.

My score is actually between weak rejection and weak acceptance. I will consider increasing my score to weak acceptance if experiments can be improved.

---post rebuttal

The new experimental results in the revised paper have addressed my concerns so I choose to raise the score to 6.

---

> ### Author Response · Authors · 2021-11-23
> **Response to Reviewer cTj2  (1/2)**
>
> Thank you for the detailed comments and valuable questions! Next we provide details to clarify your major concerns..
>
> > Q1. Novelty of this work.
>
> A1. We thank the reviewer, and will clarify our technical novelty and significance during revision.
>
> (1) First, as the first attempt on condensing graph datasets, it is **nontrivial** to develop the dataset condensation framework that shifts from images to graphs, since most components need to be adapted:
> * Modeling the dependencies between nodes.
>   * Instead of directly learning A’ and X’ as free parameters,  we show that *parameterization of graph structure and node features* can better handle the dependencies between nodes and achieve up to 8% improvement as shown in our ablation study in Table 7.  Moreover, *such parameterization better scales*, because the number of parameters does not grow quadratically with the number of condensed nodes.
>
>   * The *optimization scheme* also matters, as shown in Table 8. We turn to training the graph structure and node features alternatively. It resembles the real scenarios where node features affect the connectivities and then the connectivities also affect the node features in turn.
> * We also design the *sampling strategies* to enable efficient condensation and a *sparsification strategy* to further reduce the storage of condensed graphs.
> * Our empirical performances are very significant. We are able to approximate the original test accuracy by 99% on three datasets, 95% on one dataset and 92% on the remaining dataset, with more than 99% reduction of graph size.
>
> All the aforementioned points are **unique in the graph domain** and not necessarily straightforward to adapt from the image domain.
>
> (2) Second, condensed datasets can be **a useful point of investigation for the graph ML community**. We will open source the learned condensed datasets for further investigation. Particularly, our work yields several interesting followup directions and questions:
> * It may reveal insights into interpretability and the nature of sample-efficient representations. For example, as shown in Table 5 and Figure 2, the condensed graph provides a plausible interpretation of the original dataset and in some cases it preserves the homophily property of the original dataset, though not always.
> * Understanding the importance of preserving these properties (or good performance albeit not preserving them) is an interesting area of further research.
> * The condensed graphs can be used in many critical applications especially for neural architecture search (NAS) and continual learning. In the NAS experiments in Appendix B.2, we observe reliable correlation of performances between condensed dataset training and whole-dataset training.
>
> We believe this is an interesting and insightful work and hope you could reevaluate the contributions of our work.
>
> > Q2 Parametrization comparison. The authors claim that it is important to parameterize A' as a function of X'. However, experiments are not conducted to support this very important claim. What if A' and X' are treated as free parameters?
>
> A2.  (1) Indeed, understanding the parameterization advantages and disadvantages is important.  In fact, one of our contributions is in the proposed parameterization.  We had previously included the parametrization comparison (for Cora, Citeseer and Ogbn-arxiv) in Table 5 of Appendix B.1. Below we also include Flickr and Reddit datasets. Parameterizing A' as a function of X' tends to show more stable performance and significantly improves the performance of learning independent A’ and X’. For example, it brings improvements of 8.4% and 4.6% on Cora and Citeseer, respectively.
>
> | Parameters | Citeseer, r=1.8% | Cora, r=2.6% | Ogbn-arxiv, r=0.25% | Flickr, r=0.5% | Reddit, r=0.05% |
> |------------|------------------|--------------|---------------------|---|---|
> | $\bf A', X'$     | 62.2±4.8         | 75.5±0.6     | 63.0±0.5            |    47.0±0.1  |   86.9±2.2
> | Ours ($\bf \Phi, X'$)       | **70.6±0.9**         | **80.1±0.6**     | **63.2±0.3**            | **47.1±0.1** | **88.0±1.8**
>
> (2) It is worth noting that there are **two more benefits** of  modeling A’ as a function of X’ over free parameters.
> * Firstly, the number of parameters for modeling graph structure no longer depends on the number of nodes, hence avoiding jointly learning $O({N'}^2)$ parameters; as a result, when $N'$ gets larger, GCond suffers less risk of overfitting.
> * Secondly, if we want to grow the synthetic graph by adding more synthetic nodes condensed from real graph, the trained $\text{MLP}_{\Phi}$ can be employed to infer the connections of new synthetic nodes, and hence we only need to learn their features.

---

> > ### Author Response · Authors · 2021-11-23
> > **Response to Reviewer cTj2  (2/2)**
> >
> > > Q3. Experiments use GNNs with 2 layers. What if more layers are used? Intuitively, using fewer layers (e.g., 0 layer in the extreme case) will reduce this problem to the normal data condensation problems on non-graph data. What if a deeper GNN is used so that the graph structure plays a more important role?
> >
> > A3. Per your suggestions, we include the experiments of using more layers on the Cora dataset in Table 12 in Appendix B.5. We also show them in the table below. Specifically, we use SGC of different depth in the condensation to generate condensed graphs and then use them to test on SGC of different depth. From the table, we can see that using a deeper GNN is not always more helpful and 2-layer GNN is already good enough to produce highly generalizable condensed graphs. We conjecture that when we stack more GNN layers, more nodes are involved during the forward process, and this makes the optimization of condensed graph more difficult. As a result, we observe certain deterioration of performance on the graphs condensed with higher-layer SGC.
> >
> > | C\T | 2     | 3     | 4     | 5     | 6     |
> > |-----|-------|-------|-------|-------|-------|
> > | 2   | 80.30 | 80.70 | 79.46 | 76.06 | 71.23 |
> > | 3   |58.82 | 74.46 | 62.11 | 68.09 | 60.02|
> > | 4   | 74.24 | 72.56 | 76.26 | 71.70 | 65.12 |
> > | 5   | 71.31 | 75.73 | 70.95 | 73.13 | 67.12 |
> > | 6   | 75.20 | 75.18 | 75.67 | 76.16 | 75.00 |
> >
> >
> > > Q4. Some statements are not clear enough. For example, for the $\boldsymbol{\theta}_t$ in equation (5), it only mentions this replaces the $\boldsymbol{\theta}_t$ and $\boldsymbol{\theta}_t$  (I believe there is a typo under equation (5), too). However, how it is computed is not mentioned. Taking the average of  $\boldsymbol{\theta}_t^{\mathcal{S}}$ and $\boldsymbol{\theta}_t^{\mathcal{T}}$ ?
> >
> > A4. (1) Thank you for pointing it out. We have fixed the typo and updated the description of $\boldsymbol{\theta}_t$ in Section 3.1.
> >
> > (2) We do not take the average of $\boldsymbol{\theta}_t^{\mathcal{S}}$ and $\boldsymbol{\theta}_t^{\mathcal{T}}$. Instead, we only use *one GNN*  which is parameterized by $\boldsymbol{\theta}_t$.
> >
> > Hence, it indicates that we apply *the  same  GNN*  on  both  the synthetic, condensed  graph and the original, real graph and obtain the corresponding gradients. Then the gradients are used to calculate the matching loss. This process can be seen more clearly from Algorithm 1 in Appendix C.  After we match the gradients, $\boldsymbol{\theta}_{t+1}$ is obtained by training the GNN on the condensed graph. We have updated the description in Section 3.1.

---

### Author Response · Authors · 2021-11-28
**Summary of the major revision**

We thank the reviewers for the thorough and detailed reviews on our submission.  We summarize major changes that we have made below. All changes are marked in blue in the updated submission.

* We performed experiments on neural architecture search (NAS) in Appendix B.2 and observed reliable correlation of performances between condensed dataset training and whole-dataset training.
* We added time complexity analysis and experiments for running time in Appendix B.3 and Table 10.
* We added cross-depth experiments, which include deeper GNNs, in Appendix B.5 and Table 12.
* We included the performance comparison on Pubmed in Appendix B.7 and Table 15.
* We show the performances of various GNNs on original graphs in Appendix B.6 and Table 14 to serve as references.

---

### Decision · Program_Chairs · 2022-01-20

**Decision:**

Accept (Poster)

**Comment:**

This paper addresses the scale issue in Graph Neural Networks by proposing a “condensation” approach that produces a small synthetic graph from a large original graph such that GNNs trained on both graphs have comparable performance.

Reviewer cTj2 had concerns with novelty: they claimed the proposed method was close to gradient matching. However, they admitted that the graph setting was new. They suggested some clarity and experimental improvements.

Reviewer R5cV made a similar comment w.r.t. the similarity to gradient matching. Though overall they were more positive than R5cV and thought the idea was interesting and results were compelling.

Reviewer XqrK like the others, argued that the paper “lacked technical innovation in terms of technical contribution”. They pressed for a complexity or runtime analysis.

Reviewer peGb found the idea and problem “intriguing” though felt the solution fell short. They offered many suggestions for improving the quality of the experiments and the analysis.

In the discussion period, reviewer peGb raised their score, thanking the authors for answering their questions. They felt that the additional experiment for NAS was relevant and cleared up a key doubt. Reviewer cTj2 updated their score as well but stated it was critical that the author release the code for reproducing the new experimental results. I think that with most reviewers now on the “accept” side of the fence, I am more inclined to recommend acceptance because I do not see any critical flaws. I think that cTj2’s request for code is reasonable and strongly suggest that the authors do so.